

# The impact of mineral dust on cloud formation during the Saharan dust event in April 2014 over Europe

Michael Weger[1], Bernd Heinold[1], Ina Tegen[1], Christa Engler[2,3], Patric Seifert[1], Holger Baars[1], Fabian Senf[1], Corinna Hoose[4], Romy Ullrich[4], Axel Seifert[5], Ulrich Blahak[5], Martina Krämer[6], Ulrich Schumann[7], Christiane Voigt[7,8], and Stephan Borrmann[8,9]

[1]Leibniz Institute for Tropospheric Research, Leipzig, Germany
[2]Leipzig Institute for Meteorology, University of Leipzig, Leipzig, Germany
[3]Formerly at Leibniz Institute for Tropospheric Research, Leipzig, Germany
[4]Institute of Meteorology and Climate Research, Karlsruhe Institute of Technology, Karlsruhe, Germany
[5]Deutscher Wetterdienst, Offenbach, Germany
[6]Forschungszentrum Jülich, Jülich, Germany
[7]Deutsches Zentrum für Luft- und Raumfahrt, Institut für Physik der Atmosphäre, Oberpfaffenhofen, Germany
[8]Johannes Gutenberg-Universität, Mainz, Germany
[9]Max-Planck-Institut für Chemie, Mainz, Germany

*Correspondence to:* Bernd Heinold
heinold@tropos.de

**Abstract.** A regional modeling study on the impact of desert dust on cloud formation is presented for a major Saharan dust outbreak over Europe from 2 April to 5 April 2014. The dust event coincided with an extensive and dense cirrus cloud layer, suggesting an influence of dust on atmospheric ice nucleation. Using interactive simulation with the regional

dust model COSMO-MUSCAT, we investigate cloud and precipitation representation in the model and test the sensitivity of cloud parameters to dust-cloud and dust-radiation interactions of the simulated dust plume. We evaluate model results with ground-based and space-borne remote sensings of aerosol and cloud properties, as well as the in situ measurements obtained during the ML-CIRRUS aircraft campaign. A run of the model

with single-moment bulk microphysics without online dust feedback considerably underestimated cirrus cloud cover over Germany in the comparison with infrared satellite imagery. This was also reflected in simulated upper-tropospheric ice water content (IWC), which accounted only for 20 % of the observed values. The interactive dust simulation with COSMO-MUSCAT, including a two-moment bulk microphysics scheme and dust-cloud as

well as dust-radiation feedback, in contrast, led to significant improvements. The modeled cirrus cloud cover and IWC were by at least a factor of two higher in the relevant altitudes compared to the non-interactive model run. We attributed these improvements mainly





to enhanced deposition freezing in response to the high mineral dust concentrations. This was corroborated further in a significant decrease in ice particle radii towards more realistic

values, as compared to in situ measurements from the ML-CIRRUS aircraft campaign. By testing different empirical ice nucleation parameterizations, we further demonstrate that remaining uncertainties in the ice nucleating properties of mineral dust affect the model performance at least as significantly as to whether including the online representation of the mineral distribution. Dust-radiation interactions played a secondary role for cirrus

cloud formation, but contributed to a more realistic representation of precipitation by suppressing moist convection in southern Germany. In addition, a too low specific humidity in the 7 to 10 km altitude range in the boundary conditions was identified as a main reason of misrepresentation of cirrus clouds in this model study.

## 1  Introduction

The Mediterranean and Europe are frequently affected by outbreaks of mineral dust, as specific atmospheric circulation patterns over northern Africa and the Mediterranean cause wind-driven dust emissions over the Sahara and consecutive transport to the north (e.g. Barkan et al., 2005; Salvador et al., 2014). Estimates of annual North African dust emissions range from 400 to 2200 Tg (Huneeus et al., 2011), of which about 10 % are exported

to Europe (Shao et al., 2011). Mineral dust is an important aerosol constituent (Carslaw et al., 2010), which influences atmospheric processes. The dust particles scatter and absorb solar radiation as well as absorb and re-emit terrestrial radiation (e.g. Müller et al., 2011; Köhler, 2017), which alters the atmospheric stratification and thus can also impact cloud and precipitation formation (Chaboureau et al., 2011; Wang et al., 2013). Moreover, mineral

dust particles directly participate in cloud microphysical processes by acting potentially as cloud condensation nuclei (CCN) (Bégue et al., 2015; Karydis et al., 2011) and ice nucleating particles (INP) (DeMott et al., 2003, 2010; Boose et al., 2016).

Based on numerous field and laboratory experiments, a variety of empirical relations to describe the ice nucleating properties of mineral dust for application in numerical weather

prediction (NWP) models have been developed so far (e.g. Phillips et al., 2008; Niemand et al., 2012; Hiranuma et al., 2014; DeMott et al., 2015; Ullrich et al., 2017). The impact of dust particles on cloud microphysical and macrophysical properties cannot be generalized as it depends on the cloud type considered, the background aerosol composition, and meteorological conditions. In mixed-phase clouds, mid-tropospheric aerosol entrainment

is important to be considered (Fridlind et al., 2004), and additional INPs likely accelerate cloud glaciation, precipitation formation and finally shorten cloud life time (DeMott et al., 2010).



Cirrus clouds either form by lifting of liquid or mixed phase clouds across the homogeneous freezing threshold of 235 K (liquid origin) or in situ by a combination of heterogeneous and homogeneous ice nucleation of super cooled liquid aerosol (in situ origin) (Luebke et al., 2016; Krämer et al., 2016). If homogeneous is nucleation is primarily involved in the formation of in situ origin cirrus, ice particle concentrations are determined by this process, with a negative correlation between INP concentrations and ice particle concentrations, cloud albedo and emissivity (negative Twomey effect, e.g. Kärcher and Lohmann, 2003). If, however, lifting occurs at low vertical velocities, supersaturation over ice may never exceed the threshold for homogeneous freezing. In this case, ice nucleation is determined by deposition freezing of INPs, with the occurrence of the positive Twomey effect in in situ origin cirrus (Krämer et al., 2016).

As a result of the various atmospheric interaction modes of dust particles, the weather is likely affected by outbreaks of Saharan dust over Europe. It is, e.g., shown by observations that the efficiency of ice formation as well as the ice water content (IWC) are strongly correlated to the presence of mineral dust (Seifert et al., 2010; Zhang et al., 2018; Zhao et al., 2018). Moreover, considering ice nucleation of dust and black carbon in the GCE (Goddard Cumulus Ensemble) model, Lee and Penner (2010) found ice particle number concentrations and ice water path (IWP) increasing with higher INP concentrations for cirrus clouds. Dust particle number concentrations can exceed the hundred fold of the climatological mean value over a wide tropospheric height range during a dust event (Hande et al., 2015). In most operational NWP models, however, aerosol interactions are parameterized using preset aerosol concentrations and characteristics (e.g. The Integrated forecast system (IFS) radiation scheme uses aerosol climatology from Tegen et al. (1997), and cloud droplet and ice particle number concentrations are predefined according to an assumed aerosol background, ECMWF, 2017). Obviously, these models are challenged during those outbreaks and the forecast performance is found to be significantly reduced in the presence of mineral dust (Schumann et al., 2016). In the past, studies with interactive dust modeling approaches were conducted to quantify the effects of desert dust on weather. Smoydzin et al. (2012) included cloud activation and ice nucleation of mineral dust (diagnostically by the DeMott et al. (2010) parameterization) in the coupled chemistry model WRF-chem (Weather Research and Forecasting model – Chemistry) to simulate eastern Mediterranean dust outbreaks. Bangert et al. (2012) used the more detailed ice nucleation scheme by Barahona and Nenes (2009) with INP properties from Phillips et al. (2008) to include the competition of heterogeneous and homogeneous ice nucleation for cirrus cloud formation in their simulations of a major dust outbreak over Europe in 2008 with the regional dust model COSMO-ART (Consortium for Small-scale Modeling – Aerosols and Reactive Trace gases). Both studies found changes in mixed phase cloud microphysics due to mineral dust to



various degrees, e.g. more efficient cloud glaciation and a decrease in ice particle radii.
Using ICON-ART (Icosahedral Nonhydrostatic – Aerosols and Reactive Trace gases) with
a similar setup as Bangert et al. (2012), Rieger et al. (2017) modeled the dust outbreak over
Europe in early April 2014 in order to estimated the considerably negative impact of dust-
radiation, dust-cloud and combined effects on photovoltaic power generation.

The April 2014 Saharan dust outbreak is also the subject of this modeling study. During this
event, various cloud systems were present, but most notably, an unusually extensive cirrus
canopy occurred. The coincidence of these cloud conditions with the dust plume make it
an interesting case to investigate the impact of mineral dust on cloud formation. For the
invetigation, we use interactive regional dust transport modeling with COSMO-MUSCAT
(Consortium for Small-scale Modeling – MUltiScale Chemistry Aerosol Transport) (Wolke
et al., 2004, 2012). Particular focus is put on the treatment of heterogeneous ice nucleation
of mineral dust. Specifically we investigate: [1] how well cloudiness and precipitation are
represented in the COSMO model with the operational radiation and single-moment bulk
microphyiscs parameterizations without considering dust feedback, [2] whether consider-
ing dust-cloud and dust-radiation interactions with a two-moment microphysics scheme
improves cloud and precipitation representation, and if so, [3] how important is the role of
isolated interaction processes therein, and [4] how the choice of the INP-parameterization
influences the model results. Based on the answers to these questions, we further seek to
improve our understanding of cloud formation during the Saharan mineral dust event. We
use a comprehensive observational data set for model evaluation. It consists of standard
satellite and ground-based remote sensing, and the unique, rich data set of the campaign
ML-CIRRUS (Voigt et al., 2017), consisting of airborne in situ measurements. The present
study thus expands the work of Rieger et al. (2017), as it puts the focus on a detailed eval-
uation of cloud properties during the dust outbreak.

The paper is structured as follows: in Section 2, the interactive dust-transport model COSMO-
MUSCAT is described together with the setup of sensitivity model runs, and an overview
of the observational data available for evaluation is given. Section 3 contains a synoptic
overview of the Saharan desert dust outbreak in April 2014. In Section 4, the model results
are presented in comparison with the available observational data and a more detailed dis-
cussion of the dust impact on cloud microphysics and cloud development is given. Finally,
in Section 5 the main outcomes of the study are summarized, followed by the conclusion.





## 2 Methodology

### 2.1 Model description

For the simulations of dust transport and the effects on cloud development, the chemistry transport model MUSCAT (Wolke et al., 2004, 2012) is used, online-coupled to the non-hydrostatic regional NWP model COSMO; version 5.0, of the German Weather Service (DWD) (Doms and Baldauf, 2015).

#### 2.1.1 Operational model configuration

A detailed description of the physical parameterizations applied in the operational version of COSMO can be found in Doms (2008) and Doms et al. (2011). For the treatment of cloud processes and precipitation formation an efficient single-moment bulk water-continuity scheme is used, which considers cloud water, rain, cloud ice, snow, and optionally graupel (not used here) as hydrometeor classes. Conversion processes between these classes, as well as cloud condensation and ice formation are formulated by simple and efficient parameterizations, which do not account explicitly for the impact of a quantifiable aerosol concentration on these processes (i.e. they assume the ubiquitous presence of aerosol particles). As a result, cloud condensation and cloud evaporation is treated by performing saturation adjustment, which is the redistribution of the equivalent amount of water to restore thermodynamic equilibrium between liquid water and water vapour. This approach is reasonable for warm clouds. In mixed phase clouds, however, ice nucleation and ice particle growth occurs outside thermodynamic equilibrium, and both processes are therefore parameterized in more detail in COSMO. The underlying assumption therein is an empirical relationship between the ice particle number concentration $n_i$ and the temperature $T$, which is a fit to aircraft data from Hobbs and Rangno (1985) and Meyers et al. (1992):

$$n_i = 1 \times 10^2 \, \mathrm{m}^{-3} \exp\left[0.2 \left(T - 273.15 \, \mathrm{K}\right)\right]. \tag{1}$$

Equation 1 is used to diagnose $n_i$ as well as the mean diameter $D_i$ in the growth equation (e.g. Pruppacher and Klett, 2010) for depositional growth and to deduce an ice nucleation rate for grid cells not containing any cloud ice ($q_i = 0$):

$$\dot{q}_{i,nuc} = \frac{n_i m_i^0}{\rho_{air} \Delta t}. \tag{2}$$

$\dot{q}_{i,nuc}$ is the mass mixing ratio transferred from the water vapor to the ice phase due to heterogeneous ice nucleation per time step $\Delta t$ and involves the assumption of an initial ice particle mass $m_i^0 = 1 \times 10^{-12} \, \mathrm{kg}$. $\rho_{air}$ is the density of air. Equation 2 is only applied if the





grid cell temperature is lower than the onset temperature for ice formation $T_{nuc} = 267.15\,\mathrm{K}$. Deposition freezing is limited to temperatures lower than $T_d = 248.15\,\mathrm{K}$, whereas for temperatures above $T_d$, heterogeneous ice nucleation is the result of condensation freezing, which additionally requires water saturation.

Radiative transfer in COSMO is treated by a $\delta$-two-stream scheme, calculating upward and downward shortwave and longwave fluxes in 3 and 5 spectral intervals, respectively (Ritter and Geleyn, 1992). To consider the effects of clouds on radiative transfer, a cloud fraction is parameterized, which encompassed contributions from grid-scale and sub-grid scale stratiform cloudiness as well as convective cloudiness. Accordingly, modified liquid and ice water mixing ratios ($q_{sc,c}$ and $q_{sc,i}$), containing the sub-grid scale contributions, are derived, which are used to calculate optical properties of clouds. Most importantly, the generalized effective diameter $D^{ef}$ is directly related to $q_{sc,c/i}$ via empirical formulations. Radiative transfer further depends on the vertical alignment of cloud free and cloud covered areas in adjacent layers. It is assumed that clouds have maximum overlap, unless there is an intermediate layer without any cloudiness. In this case clouds are distributed randomly. To include the effects of aerosols, a spatially variable climatological mean aerosol distribution is prescribed, with consideration of 5 different types of aerosol optical properties (maritime, continental, urban, volcanic and background stratospheric).

### 2.1.2 Dust scheme

Dust emission and transport are computed by the multiscale transport model MUSCAT, including the parameterization of dust emission and deposition fluxes is given in Heinold et al. (2007), Heinold et al. (2011) and Schepanski et al. (2017). Mineral dust is transported as a passive tracer in five size bins with the particle diameter limits at $0.2\,\mu m$, $0.6\,\mu m$, $1.8\,\mu m$, $5.2\,\mu m$, $16\,\mu m$ and $48\,\mu m$. For dust advection, in MUSCAT, a third-order upstream scheme is used along with an implicit–explicit integration scheme (Knoth and Wolke, 1998; Wolke et al., 2000). The dust source scheme is based on the work of Tegen et al. (2002) and includes the parameterization of the threshold friction velocity $u_t^*$ for particle mobilization. $u_t^*$ is dependent on the soil particle size distribution (Marticorena and Bergametti, 1995), which is resolved in four size classes (coarse sand, medium/fine sand, silt and clay) and the surface roughness length $z_0$. To account for the effect of vegetation on dust emission, 27 different vegetation types are considered. Vegetation cover is further parameterized according to Knorr and Heimann (1995), using satellite based normalized difference vegetation index (NDVI) data sets (Tucker et al., 2005). Based on the vegetation type and cover as well as snow cover, an effective area $A^{ef}$ for dust emission is calculated. Soil moisture content, derived from the hydrological fields of COSMO, is assumed to suppress dust emission, if exceeding 99 %. The frictional velocity $u_*$ is calculated from COSMO first layer winds us-





ing surface roughness data from satellite retrievals (Prigent et al., 2012). If $u_* > u_t^*$, dust
emission is allowed and computed with a cubic function of $u_*$ (Heinold et al., 2007). The
potential areas of dust emission are prescribed using a dust source activation frequency
mask. This is derived from Meteosat Second Generation (MSG) Spinning Spinning En-
hanced Visible and InfraRed Imager (SEVIRI) dust index observations (Schepanski et al.,
2017). Dust removal is treated as dry (Zhang et al., 2001) and wet deposition, while the
latter considers in-cloud and below cloud scavenging (Berge, 1993; Jacobson, 1997; Jonson
et al., 1998).

Comparisons with results from field studies show that the model provides a good repre-
sentation of the different aspects of the atmospheric dust cycle (e.g. Heinold et al., 2011).

### 2.1.3 Dust-cloud interactions

For the interaction of simulated dust with clouds, the two-moment bulk microphysics
scheme of Seifert and Beheng (2006) as also implemented in COSMO was modified to in-
clude the effects of a variable mineral dust concentration on cloud activation and hetero-
geneous ice nucleation. In past modeling studies, it was commonly assumed for in-cloud
conditions that all aerosol particles are scavenged by the cloud droplets (e.g., Bangert et al.,
2011). As a consequence, cloud freezing had to be treated stochastically only depending on
cloud droplet number concentrations but not on a variable aerosol concentrations (Bangert
et al., 2012). Field studies, however, have shown the assumption of complete in-cloud scav-
enging is usually violated, especially toward the cloud edges (Gillani et al., 1995) or in the
presence of ice particles (Verheggen et al., 2007). In this work, for a detailed description
of in-cloud droplet activation as well as ice nucleation (immersion, contact and deposi-
tion nucleation), the aerosol concentration $n_a$ is partitioned into an interstitial $n_{in}$ and by
cloud water scavenged component $n_{sc}$. Aerosol species considered are mineral dust, soot
and organics, while the latter are given by prescribed number concentrations (see Tab. 2).
The cloud number concentration $n_c$ is used to determine $n_{sc}$, and accordingly if $n_c > n_a$,
$n_{sc} = n_a$, otherwise $n_{sc} = n_c$. In latter case, the 5 particle size classes are partitioned by
applying weighting factors based on data of a case study report by Hallberg et al. (1994),
which prioritize the larger particle classes.

Cloud droplet activation is parameterized according to AbdulRazzak and Ghan (2000) for
a multi-mode aerosol, consisting of different size classes and different chemical groups.
The different chemical composition of dust, soot and organics is represented by different
hygroscopicity parameters $\kappa$. We use the following set of hygroscopicity parameters for
dust, soot and organics: 0.14, 0.308, 0.308, respectively. The parameterization was orig-
inally developed for droplet activation at the cloud base, considering the competition of
the different aerosol modes in an ascending air parcel. To modify the parameterization for




in-cloud conditions, preexisting cloud droplets are considered as an additional competing
aerosol mode with the size being the mean droplet diameter $D_c$ and with $\kappa \approx 0$. Conse-
quently, only the interstitial aerosol component is available for droplet activation.

Heterogeneous ice nucleation in our model is based on empirical parameterizations of the
aerosol surface density of ice nucleation active sites (INAS) $n_{IS}$ [μm$^{-2}$], presuming the va-
lidity of the singular hypothesis. For desert dust, we use the parameterization of Ullrich
et al. (2017) as default, which can be considered as one of the most accurate to date. It is
based on a comprehensive data set gathered by nucleation experiments, carried out in the
Aerosol Interaction and Dynamics in the Atmosphere (AIDA) facility (Wagner et al., 2006),
and a novel algorithm for data evaluation. It is supposed to be an especially reliable param-
eterization for deposition nucleation as it shows the characteristic u-shape of INAS-density
isolines, which is in accordance with more recent theoretical work on deposition nucleation
(Marcolli, 2014). For soot and organics, we use the parameterization of Phillips et al. (2008),
which is based on field studies.

Heterogeneous cloud droplet freezing is determined by the probability $P_{fr}$ for a single
cloud droplet to freeze, hence after evolution of model time step $\Delta t$, the number of het-
erogeneously frozen droplets is:

$$\Delta n_{c,het} = -n_c P_{fr}. \tag{3}$$

$P_{fr}$ results from the combined probabilities for immersion ($P_{im}$) and contact freezing ($P_{co}$):

$$P_{fr} = 1 - (1 - P_{im})(1 - P_{co}). \tag{4}$$

In a first order approximation, $P_{co}$ is proportional to the number of colliding interstitial
aerosol particles with cloud droplets during evolution of model time step $\Delta t$. To parame-
terize the collision rate, the collision kernel $\Psi_{co}^l$ of Ovtchinnikov and Kogan (2000) is used,
which includes the attractive or repulsive forces of Brownian motion, thermophoresis and
diffusiophoresis. It depends on the diameter of colliding particles, approximated here as
the first moment of the cloud droplet PSD and the mean diameter of considered aerosol
particle size bin $l$. In Eq. 5, it is summed over all aerosol indices for chemical class k and
size l.

$$P_{co} = \sum_{k,l} n_{in}^{k,l} \Psi_{co}^l P_{IN}^{k,l} \Delta t. \tag{5}$$

$P_{im}$ results from immersed potential INPs, which activate if the temperature tendency
$\Delta T = T(t + \Delta t) - T(t)$ is negative, which leads to a temporal increase in INP concentra-





tions. $\Delta T$ is diagnosed using the grid-scale vertical velocity $w$ and the vertical temperature gradient $\mathrm{d}T/\mathrm{d}z$, thus neglecting horizontal temperature advection. Only the scavenged aerosol component $n_{sc}$ is available for immersion freezing:

$$P_{im} = \frac{1}{n_c} \sum_{k,l} n_{sc}^{k,l} \Delta P_{IN}^{k,l}(T, \Delta T_t),$$  (6)

Both Eq. 5 and 6 contain the probability $P_{IN}$ of an aerosol particle to act as an INP, which is based on a Poisson distribution:

$$P_{IN}^{k,l}(T) = 1 - \exp\left\{-n_{IS}\left[T, S_i^w(T)\right]\omega^{k,l}\right\}.$$  (7)

Therein the expectation value refers to the mean number of INAS per particle being ac-
tive at temperature $T$ and saturation over ice $S_i$. $\omega^{k,l}$ is the mean aerosol surface, of the considered aerosol mode. For contact and immersion freezing, $n_{IS}$ is evaluated at water saturation, as indicated by $S_i = S_i^w(T)$. The freezing threshold for contact freezing was found to be about $4.5\,\mathrm{K}$ higher than for immersion freezing (Shaw et al., 2005), thus in the case of contact freezing a correction term $\Delta T_{co} = 4.5\,\mathrm{K}$ is applied in the calculation of $n_{IS}$.
For contact freezing, colliding aerosol particles are represented by a population of com-pletely inactivated aerosol particles. For immersion freezing, however, the fraction of al-ready frozen INPs needs to be taken into account by calculating the increase of $P_I N$ during $\Delta t$:

$$\Delta P_{IN}(T, \Delta T_t) = \mathrm{Max}\left[0, P_{IN}(T + \Delta T) - P_{IN}(T)\right].$$  (8)

Deposition freezing of water vapour on interstitial aerosol particles predominantly takes place in pure ice clouds at $S_i > 1$ and $T < 235\,\mathrm{K}$. These restrictions are not explicitly made here, as the empirical INP-parameterizations for deposition freezing are also valid for higher temperatures. In most cases, however, cloud droplet freezing and deposition nu-cleation are not expected to occur simultaneously at significant rates, as at water saturation
and for $T > 235\,\mathrm{K}$ deposition nucleation is not efficient. The number of newly nucleated ice particles in the deposition freezing mode $\Delta n_{i,dep}$ is calculated diagnostically according to Seifert and Beheng (2006), as a balance equation for interstitial aerosol would be needed for a prognostical treatment (in opposition to cloud droplet freezing, where there is a balance equation for $n_c$). Thus, $\Delta n_{i,dep}$ is limited by the ice and snow particle number concentra-
tions $n_i$ and $n_s$, respectively:

$$\Delta n_{i,dep} = \mathrm{Max}(0, \sum_{k,l} n_{in}^{k,l} P_{IN}^{k,l} - n_i - n_s).$$  (9)





Finally, the number of heterogeneously frozen ice crystals due to ice nucleation $\Delta n_{i,het}$ is given as the sum of heterogeneous cloud droplet freezing and deposition freezing:

$$\Delta n_{i,het} = -\Delta n_{c,het} + \Delta n_{i,dep} \tag{10}$$

Homogeneous freezing of cloud droplets is treated as in Seifert and Beheng (2006), with a stochastic approach and a temperature dependent freezing rate constant (Cotton and Field, 2002).

### 2.1.4 Dust-radiation interactions

The computation of short- and longwave radiative fluxes in COSMO considers scattering,
absorption and re-emission by aerosols, cloud hydrometeors and trace gases. In interactive COSMO-MUSCAT simulations, it additionally takes into account the modeled size-resolved dust distribution (Helmert et al., 2007). The model thus considers the direct radiative impact and related dynamical feedbacks of the spatially and temporally varying atmospheric dust load. Dust optical thickness is calculated based on the modeled dust
concentration by assuming spherical particles. The optical properties of Saharan dust are derived from Mie theory (Mishchenko et al., 2002) using refractive indices from Sinyuk et al. (2003).

In order to make use of the more detailed two-moment microphysics information, the single-moment microphysics approach of parameterizing cloud optical properties in Rit-
ter and Geleyn (1992) was revised by Dipu et al. (2017). This way also the impact of the spatially and temporally varying size of cloud droplets and ice crystals on cloud optical depth and reflectivity is accounted for, by using the prognostic version of the effective diameter $D^{ef}$ of cloud droplets and ice crystals. For cloud ice, $D_i^{ef}$ is defined by:

$$D_i^{ef} = \frac{3\,\mathrm{IWC}}{2\rho_i \sigma_i} \tag{11}$$

(e.g. Mitchell et al., 2011), with the specific mass of ice $\rho_i$, the water content IWC and the mean particle cross-section $\sigma_i$. Assuming shape parameters for ice particles, IWC and $\sigma_i$ are directly computed from the prognostic variables $n_i$ and $q_i$ (and similarly for cloud water content LWC from $n_c$ and $q_c$), which should lead to a much more accurate representation of dust-cloud-radiation effects in the model.

### 2.2 Simulation setup

COSMO-MUSCAT is applied on two-fold nested domains, as depicted in Fig. 1, to simulate the Saharan dust outbreak in early April 2014 over Europe. The outer domain D1,





which covers Europe and North Africa provides the emission and long-range transport of Saharan dust toward Europe, while the inner domain D2 is used for investigating the representation of cloudiness and precipitation over Germany. Domain D1 has a horizontal resolution of 14 km, is divided into 40 vertical layers up to an altitude of 20 km, and spans the area enclosed within 20° N, 20° W and 61.5° N, 39° E. In this simulation, COSMO is run with the operational single-moment bulk water continuity microphysics scheme and without dust-cloud interactions. However, interactive dust-radiation interactions according to Heinold et al. (2007) were considered. A Tiedtke (Tiedke,1989) convection scheme was used to treat sub-grid scale cloud and precipitation processes, related to moist convection, as horizontal resolution is not sufficient to explicitly resolve these processes. The dust simulation is run for the period 27 March to 6 April, in order to cover associated dust emissions and the development and evolution of the dust plume. COSMO is driven by initial and boundary fields from analysis of the global GME model of DWD. The simulations are re-initialised every 48 h to keep the model meteorology close to the real synoptic situation. One cycle consists of 24 h of meteorological simulation, followed by another 24 h of COSMO-MUSCAT dust simulation. This allows enough relaxation time for the meteorological fields after re-initialization, as only the second half of the cycle is evaluated. The dust distribution of the previous run is used to initialize the following, respectively. The dependency of dust emission on surface winds is highly non-linear. The modeled dust is therefore highly sensitive to uncertainties in surface and soil properties as well as predicted wind speed. In order to match satellite and ground based observations of dust optical thickness, the threshold velocity for dust mobilization is reduced by a factor of 0.63.

For the simulation of dust-cloud interactions over central Europe, COSMO-MUSCAT is run on the inner domain D2 with 2.8 km resolution and with 50 vertical layers. The area spans the coordinate range enclosed within 48.3 ° N, 4.0 ° E and 55.3 ° N, 13.0 ° E. The simulation is started on 3 April 2014, 00:00 UTC and run for 60 hours without restart. COSMO is initialised and driven with hourly analysis data from the operational COSMO-DE run provided by DWD in order to use driving conditions that are closest to the actual weather situation and benefit from the finer grid spacing. The simulated dust fields of D1 are interpolated onto the D2 grid and used for initialisation of the dust fields of D2, as well as for the 6-hourly updated lateral boundary conditions. As the model-domain resolution permits moist deep convection, the Tiedke scheme is restricted to parameterize shallow convection only.

One D2 model run is performed with the single-moment bulk microphysics scheme as for D1. To ensure numerical stability of the microphysics schemes, the model integration time step is lowered to 10 s compared to the 25 s standard. Model evaluation is started after a



**Table 1.** Overview of the model runs performed in this study to investigate dust-cloud interactions (DCI) and dust-radiation interactions (DRI). CLM denotes the spatially and temporally fixed climatological mean dust concentration, INT indicates the interactively simulated dust concentration. U17 refers to the parameterization of Ullrich et al. (2017), and P08 to the parameterization of Phillips et al. (2008)

.

| Run | Purpose | Dust INAS density | Dust INP | Dust CCN | Dust radiation |
|------|---------|------------------|----------|----------|----------------|
| **SMBLK** | Reference, without DCI and RCI | - | - | - | - |
| **ICLM** | DCI and DRI at normal low dust conditions | U17 | CLM | CLM | CLM |
| **IINT** | DCI and DRI of simulated dust outbreak | U17 | INT | INT | INT |
| **RCLM** | Evaluate DRI of climatological dust | U17 | INT | INT | CLM |
| **CCLM** | Evaluate CCN-effect of climatological dust | U17 | INT | CLM | INT |
| **IAIP** | Test alternative ice parameterization for dust | P08 | INT | INT | INT |

model spin-up time of 24 h on 4 April 2014, 00:00 UTC, which roughly coincides with the appearance of the dense cirrus canopy over Germany in the satellite images.

**Table 2.** Aerosol size distribution for dust, soot and organics used in COSMO-MUSCAT for the climatological mean background. The number concentrations for dust are based on the temporal and spatial mean of simulated dust fields of a model run carried out on domain D2. The values for soot and organics are taken from Phillips et al. (2008), and accredited to the smallest size bin. The size bins are distributed logarithmically with a relative standard deviation of $\sigma = 2$.

| size bin | mean volume diameter [μm] | number concentration [m$^{-3}$] dust | soot | organics |
|----------|---------------------------|------|------|----------|
| **1** | 0.39 | $1.10 \times 10^5$ | $1.50 \times 10^7$ | $1.77 \times 10^8$ |
| **2** | 1.17 | $5.02 \times 10^4$ | - | - |
| **3** | 3.53 | $2.00 \times 10^3$ | - | - |
| **4** | 10.65 | $2.18 \times 10^2$ | - | - |
| **5** | 32.16 | $2.36 \times 10^{-6}$ | - | - |

  To evaluate the effects of the dust plume on cloud activation, ice nucleation and radiation, four additional model runs are carried out on D2, deploying the two-moment scheme by Seifert and Beheng (2006) with the modifications to allow for the online feedback of

360 dust on cloud activation and ice nucleation. For ice nucleation of desert dust, the most up-



to-date INAS density parameterization of Ullrich et al. (2017) (U17) is chosen as default, while for the climatological background aerosol of soot and organics we rely on the parameterization of Phillips et al. (2008) (P08). A summary of the sensitivity model runs is given in Tab. 1, and parameters of the climatological mean aerosol particle size distribution (PSD) are listed in Tab. 2. The run ICLM is used to represent a climatological background dust scenario with two-moment microphysics, while IINT uses the simulated dust fields instead of the constant prescribed dust PSD. In RCLM and CCLM dust-radiation and dust-cloud-activation are computed using the prescribed climatological mean dust concentration (modeled dust PSD, but with dust amount reduced to spatially uniform low average value), respectively, in order to disentangle those effects of the simulated dust plume from the fully interactive effects seen in IINT. Finally IAIP is analogous to IINT, but uses the parameterization of P08 for ice nucleation of mineral dust.

### 2.3 Observational data

#### 2.3.1 Cloud radar observations

To evaluate the modeled IWC, 94 GHz spaceborne and 35 GHz ground-based cloud radar observations are used. From the model side, the contributions form the mixing ratios of cloud ice ($q_i$), snow ($q_s$), graupel ($q_g$), as well as the sub-grid scale ice ($q_{i,sc}$) have to be included. $q_{i,sc}$ is parameterized with a relative humidity scheme and the stratiform cloud fraction. By using the density of air, the values are converted to units $[\mathrm{g\,m^{-3}}]$.

Observational data comprise a vertical cross section of IWC along a CloudSat satellite overpass on 4 April, 12:30 UTC (granules: 12457-42209) with an along-track resolution of 1.7 km (data product 2B-CWC-RO P_R04, Austin et al., 2009). To compare these data to model results, overflown grid cells are extracted from the D2 domain and observations are interpolated onto this array. Furthermore, vertical coordinate transformation to the 51 layer boundary heights of COSMO-MUSCAT is carried out, by averaging over all observations located within the corresponding model layer. Finally, horizontal grid-cell averages are computed for both observational and model data.

In addition to the CloudSat profile, a time series of vertical profiles of IWC retrieved from the 35 GHz zenith pointing radar at Leibniz Institute for Tropospheric Research (TROPOS) (51.3° N, 12.3° E) is available. The cloud radar of type Mira-35 (Görsdorf et al., 2015) is operated within the Leipzig Aerosol and Cloud Observations System (LACROS, Bühl et al., 2013) which comprises in addition an extensive set of active and passive ground-based remote sensing instrumentation, such as lidar (PollyXT, Engelmann et al., 2016), Microwave radiometer (HATPRO, Rose et al., 2005) and optical disdrometer. The observations of LACROS are automatically processed within Cloudnet (Illingworth et al., 2007) based



on which a hydrometeor and aerosol target categorization is derived. Cloudnet provides output with a temporal and vertical resolution of 30 s and 30 m, respectively. The Cloudnet target classification builds the basis for the retrieval of products such as liquid water content and ice water content. The ice water content is derived for all identified ice-only

measurement points based on a parameterization of Hogan et al. (2006) that uses an empirical relationship between ice water content, temperature, and radar reflectivity factor. In order to compare these data to equivalent model results, in a first step the original data set is averaged over variable time periods around the dates of the model output with 15 min intervals. The number of measurements to include in a single averaging procedure is given

by horizontal advection and is therefore calculated based on the horizontal grid spacing of 2.8 km and the modeled horizontal wind speed. After vertical coordinate transformation, the obtained data sets are time averaged over the period from 4 April, 00:00 UTC to 5 April, 12:00 UTC.

### 2.3.2   Infrared satellite imagery

Model output fields of the hydrometeor mixing ratios, as well as thermodynamic variables are supplied to an infrared (IR) forward simulation (see Appendix A for a detailed description). Resulting synthetic infrared images can be compared to satellite images obtained with the Spinning Enhanced Visible and Infrared Imager (SEVIRI) instrument aboard Meteosat Second Generation (MSG) satellite and provided by EUMETSAT (https://www.

eumetsat.int). For the atmospheric window channel at 8.7 µm, brightness temperature is used as a proxy for cloud top temperature of optically dense clouds, and further as a measure of cloud top height.

### 2.3.3   Precipitation records

For Germany and closely surrounding areas, hourly precipitation totals are available for a

total of 970 stations. The data is provided by the Climate Data Center (CDC) of the German weather service DWD (http://www.dwd.de/cdc). The data was integrated over the model evaluation period from 4 April 2014, 00:00 UTC to 5 April 2014, 12:00 UTC and is used to evaluate the modeled precipitation amount for the different model runs.

### 2.3.4   ML-CIRRUS cloud microphysical measurements

During the ML-CIRRUS campaign (Voigt et al., 2017), 16 flights were performed from 26 March to 15 April 2014 with the High Altitude and Long Range Research Aircraft (HALO). The campaign had the scope to investigate cirrus and contrail cirrus above Germany and Western Europe with a novel in situ and remote sensing payload. For model evaluation, the flights conducted on 3 and 4 April provide valuable information on cloud microphysical



and thermodynamic properties.

Cloud particle number concentrations for 3 April were measured with the particle spectrometer NIXE-CAPS (Baumgardner et al., 2001; Meyer, 2013), which consists of the cloud and aerosol spectrometer NIXE-CAS to measure size and concentration of particles in the diameter range between 0.61 µm to 50 µm, and the optical particle counter NIXE-CIP to
measure particles in the diameter range of 15 µm to 945 µm with 15 µm resolution. On 4 April, data were not available for the NIXE-CIP instrument. For particle diameters larger than 25 µm, we therefore used available measurements from the Cloud Combination Probe (CCP) CIP instrument (Weigel et al., 2016). To limit aerosol contamination, we do not consider size bins containing particles smaller than 3 µm. PSD of ice particles in the form
of particle number concentration density $[\mathrm{m^{-3}\,\mu m^{-1}}]$, if not yet calculated, is simply approximated by dividing absolute particle counts in the sample volume of $10^{-6}\,\mathrm{m^3}$, by the diameter range of each size bin. According to the aircraft altitude, measurements were assigned to the corresponding vertical layer of COSMO-MUSCAT on D2. For each layer the horizontally averaged PSD was calculated. Measurements with non-significant particle
concentrations ($n_i < 1\,\mathrm{m^{-3}}$) were not considered. IWC was retrieved from the measured PSD, assuming empirical mass-diameter relationships according to Krämer et al. (2016) and using the arithmetic mean of the size bin limits. For a measurement, the sphere-equivalent mean diameter $D_i$ can be obtained simply by:

$$D_i = \left(\frac{\mathrm{IWC}}{n_i \rho_i}\right)^{\frac{1}{3}}. \tag{12}$$

The number-averaged diameter $\overline{D}_N$ of an ensemble of measurements is then defined by:

$$\overline{D}_N = \sum D_i n_i / \sum n_i, \tag{13}$$

and similarly the IWC-averaged $\overline{D}_{\mathrm{IWC}}$ by:

$$\overline{D}_{IWC} = \sum D_i \,\mathrm{IWC} / \sum \mathrm{IWC}. \tag{14}$$

To find comparable model data, for each HALO measurement a horizontal circle with
radius $r = 100\,\mathrm{km}$ around the aircraft position is defined. The selection of this radius is justified, when considering the large impact of randomness on the distribution of clouds in the model at this length-scale. Within this circle and at the model layer in which the aircraft is situated, the closest grid cell to the aircraft position is taken, which further has a grid-cell average IWC value in the same order of magnitude than that of the respective measure-
ment. For the comparison of IWC, both measured, as well as modeled IWC, containing



contributions from $q_i$ and $q_s$, are discretized in 9 levels. If there is no grid cell meeting this criterion, the respective measurement is excluded from a direct comparison with model results. By this approach, the co-dependence of $D_i$ on IWC is taken into account, as well as to some extent the geographical co-dependence, which increases the significance of the
comparison. For different model runs, however, the resulting number of data pairs can be different, as clouds are differently distributed in the model runs. For the ML-CIRRUS flight on 4 April, this approach cannot be applied, as the HALO flight-track is located completely outside of the domain D2. Nevertheless, these observations are useful for comparison with data from the ML-CIRRUS flight conducted on 3 April.

**2.3.5 Atmospheric soundings**

To evaluate the thermodynamic state of the atmosphere in the model, 12 or 6 hourly data-sets of atmospheric soundings (source: University of Wyoming; http://weather.uwyo.edu/upperair/sounding.html) are used from the stations Essen (51.40 ° N, 6.97 ° E), Norder-ney (53.71 ° N, 7.15 ° E), Schleswig (54.53 ° N, 9.55 ° E), Greifswald (54.10 ° N, 13.40 ° E), Bergen
(52.81 ° N, 9.93 ° E), Lindenberg (52.21 ° N, 14.12 ° E), Kümmersbruck (49.43 ° N, 11.90 ° E), Meiningen (50.56 ° N, 10.38 ° E) and Idar-Oberstein (49.70 ° N, 7.33 ° E). For evaluation of the model results, vertical profiles of specific humidity are used, which is regarded as the main limiting factor for ice formation. For the dates 4 April, 00:00 UTC, 4 April, 12:00 UTC and 5 April, 00:00 UTC, vertical profiles of specific humidity for the aforementioned sta-
tions were averaged. This data set can be directly compared to corresponding model data extracted at the closest geographical positions to the stations.

**2.3.6 Dust aerosol observations**

For dust model evaluation, a comprehensive set of Aerosol Robotic Network (AERONET) (https://aeronet.gsfc.nasa.gov, Holben et al., 1998) sun photometer observations of aerosol
optical depth (AOD) are used. Measurement sites are located in northern Africa and Europe, with a total of 40 available stations over the simulation period. While sun photometer measurements are affected by all types of aerosol, COSMO-MUSCAT AODs only consider dust. However, the coarse mode AOD product (level 2.0 quality) of AERONET (O'Neill et al., 2003) predominantly represents the dust fraction and is thus commonly used for
evaluating dust only model results. In addition to the AERONET observations, a parti-cle extinction coefficient profile, retrieved from lidar measurements (Baars et al., 2016) at TROPOS in Leipzig is used to evaluate the modeled vertical mineral dust distribution over Europe. Therefore, modeled mass extinction coefficients are derived from the dust PSDs by using refractive indices of Sinyuk et al. (2003) applying Mie theory. With further vertical in-
tegration of the mass extinction coefficients, dust AOD values are obtained for comparison



with AERONET data.

## 3 Dust outbreak April 2014

In early April 2014, a pronounced trough of low pressure was situated over the eastern Atlantic Ocean, placing western Europe under a recurrent southerly flow pattern. On 2 April, the trough propagated eastward with the associated cold front reaching the Atlas Mountains at the Moroccan/Algerian border, where it initiated a small lee cyclone to the south of the mountain range. Consecutively, high surface winds caused large dust emissions in the afternoon on 2 April. Figure 2a shows the MSG IR false color dust index indicating dust presence by magenta and purple colored shadings. In addition, isolines of 500 hPa geopotential height illustrate the synoptic situation at this time. The associated intense dust plume can be clearly identified from the magenta coloring over the Moroccan/Algerian border. Furthermore, the 500 hPa geopotential height contour lines indicate air mass transport toward the western Mediterranean basin. Inside the conveyor belt, which closely goes ahead of the cold front, strong lifting caused the Saharan dust eventually to reach the upper troposphere, where it was further transported eastward behind the pronounced ridge axis. Based on this analysis, a destination of the lifted dust over western and central Europe can be expected.

Over the consecutive day, the eastward travelling cold front caused nearly continuous dust emissions (albeit weaker than the event in the afternoon on 2 April) over the desert in Algeria and Tunisia (see purple features in Fig. 2b). The persistence of the aforementioned flow pattern also favored most of this dust eventually reaching Europe on 3 April. With the development of an upper level cut-off low over the western Mediterranean Sea and with its eastward movement, upper level winds over south-central Europe became more and more easterly on 4 April and as a consequence, dust export from North Africa to central Europe was not further supported. Ongoing upper level lifting, especially at the northern boundary of the advancing Saharan mineral dust-rich airmas caused the development of extensive cirrus cloudiness. The horizontal extension as well as the high optical density of the cirrus shield became striking in the morning hours of 4 April, as associated cloud top temperatures reached below 210 K over a large area, covering parts of France, Great Britain, the Benelux States, and Germany (see Fig. 2c). A link of the high atmospheric mineral dust concentrations to this cloud development seems likely, as desert dust has excellent ice nucleating abilities and furthermore it can destabilize the thermodynamic stratification of the atmosphere at the upper edge of the dust plume through interaction with radiation. Weak anticyclonic currents over central Europe kept the dust airmass and the associated cloudiness trapped until 5 April. In the evening hour an eastward moving Atlantic cold front





finally marked the end of the dust event, as skies cleared up just behind in the replacing clean and subsiding airmass.

In the MSG imagery, cloud cover obscured most of the dust transport towards Europe. The horizontal dust distribution is better seen in the dust AOD maps (Fig. 3) from the COSMO-MUSCAT simulation at 14 km horizontal resolution. Figure 3a shows an intense dust plume with AOD exceeding 1.5 at the Moroccan/Algerian border on 2 April, 15:00 UTC, which corresponds to the magenta dust signature in the corresponding MSG IR image. Obviously, at this time, Europe was still affected by the remnants of a previous dust outbreak which occurred in late March 2014. In the consecutive image (Fig. 3b), the dust transport towards Europe is clearly depicted within the s-shaped conveyor belt. The initial dust plume is now located over Germany with AOD values still reaching up to 1.5. Furthermore, the model reproduces the significant dust emissions in terms of dust AOD over the desert in Algeria and Tunisia in association with the eastward travelling cold front. In Fig. 3c, the closed circulation centred over Sardinia is clearly seen, with the freshly emitted dust over Libya being steered increasingly in a cyclonic gyre over the Mediterranean. Meanwhile the dust over central Europe is kept trapped, with the highest dust load found over Austria and southern Germany at AOD values up to 1. On 5 April, 12:00 UTC (Fig. 3d) the dust AOD over Germany is markedly decreased but locally still reaches up to 0.5.

The available AOD observations of AERONET stations (550 nm coarse mode, quality level 2.0) are depicted by colored circles in Fig. 3 for dust model evaluation. On 2 April, 15:00 UTC, observed AOD over Europe is already significantly raised (up to 0.5), which is similar to the model results. On 3 April, 15:00 UTC, observations show a strong zonal gradient in AOD, with strongly elevated values in Tunis and Lindenberg (Germany) (AOD up to 1) and moderately elevated values in northern Germany (AOD up to 0.3), while the Iberian peninsula was obviously not affected by dust (AOD< 0.01). This is in very good agreement with the simulation. Over the next 2 days, there is a lack of observations over central Europe due to the obscuring cloudiness. However, the AERONET station at Lindenberg shows AOD values up to 0.5 on both 4 and 5 April, 12:00 UTC, which is higher than in the model, where the highest dust loads are displaced more to the south.

A statistical evaluation, taking AERONET coarse mode AOD data of level 2.0 quality and corresponding modeled values at the nearest geographical position into account, is shown by the scatter plot in Fig. 4. The considered data set contains 301 observation/model pairs collected over the period 2 April 00:00 UTC to 6 April 00:00 UTC. Correlation between observational and model data is 0.57. One has to note, however, that the number of observations directly affected by the dust plume was supposedly lower than usual, due to the extensive cloud cover. The mean AOD of both observations and model data does not differ significantly, as it is $0.07 \pm 0.06$ and $0.06 \pm 0.07$ respectively.



Concerning the vertical distribution of mineral dust, Fig. 5a and 5b show dust particle
number concentrations $n_d$ along the meridian $10°$ E, running through central Germany
from the COSMO-MUSCAT simulation at 2.8 km horizontal resolution. The plots confirm
that the Sahara dust reached the highest layers of the troposphere, with $n_d$ locally reaching
up to $1×10^8$ m$^{-3}$. This is the thousand fold of the climatological mean value of $1.6×10^5$ m$^{-3}$
as in Phillips et al. (2008). Nearly everywhere in the cross section, the values are above the
climatological mean for both 4 and 5 April, 12:00 UTC. Lower values are likely caused by
local wet deposition in association with precipitation over northern Germany.

A vertical particle extinction coefficient profile, retrieved from lidar measurements (Baars
et al., 2016) at TROPOS in Leipzig provides further valuable information for dust model
evaluation (Fig. 6). Albeit averaged for a short time period around 5 April, 21:00 UTC,
when the main dust event had already ended over most of Germany, it still shows mineral
dust up to an altitude of 7 km. Above this altitude, already clean air had subsided. According to the lidar particle extinction retrievals, the dust was located in two distinct layers.
Associated peak values are well above the hundred fold of the corresponding climatological mean value (see blue vertical line). The simulated extinction coefficient profile, which is
computed from COSMO-MUSCAT dust concentrations using refractive indices taken from
Sinyuk et al. (2003), is in good agreement, as it shows peak values of the same magnitude as
well as the strong decline of mineral dust extinction above 6 km altitude. However, the layered structure is not well reproduced. Partly, this can be attributed to black carbon aerosol
incorporation, which has very similar absorbing properties as mineral dust, in the lidar
observations, which is not considered in the dust simulations by COSMO-MUSCAT. To a
large extent, however, this is likely due to a too strong vertical mixing and the potential incorporation of anthropogenic air pollutants within the boundary layer. COSMO-MUSCAT,
on the other hand, was only applied to simulate wind-driven dust emissions from soil surfaces.

## 4 Results

### 4.1 Reference model run SMBLK

We first compare the model run SMBLK (see Tab. 1) with single-moment bulk microphysics
and without dust feedback on clouds and radiation with the available observational data
for an initial assessment of the representation of cloud cover, cloud microphysics and precipitation. For qualitative cloud cover comparison, we derived synthetic infrared satellite
images by the application of the infrared forward simulator (see Appendix A) on the model
data. Figure 7 shows maps of brightness temperature from MSG SEVIRI and the model run
SMBLK over a 24 h period, starting on 4 April, 12:00 UTC. An extensive and optically dense



shield of cirrus clouds traverses the domain in the satellite images. In some areas the associ-
ated cloud top temperatures reach below 210 K, which corresponds to the tropopause level.

Most of this cloudiness is not present in the synthetic images of the model data, where it is
mostly limited to the western and northern boundaries. These regions are strongly influ-
enced by the driving boundary fields, as winds mostly prevailed from the west and north.
Moreover, cloud top temperatures of cloud fields present in the model are about 20 K too
warm in comparison with the satellite images. In Fig. 7d and 7f, clouds are completely

missing in the southeastern parts of the domain in the model results.

Comparing the available CloudSat data to modeled IWC for the satellite overpass on 4
April around 12:30 UTC in Fig. 8a, reveals a significant lack of IWC above an altitude of
6 km in the model. The discrepancy reaches up to one order of magnitude at 10 km alti-
tude with a modeled value of $1 \times 10^{-3}\,\mathrm{g\,m^{-3}}$ compared to $1 \times 10^{-2}\,\mathrm{g\,m^{-3}}$ in the CloudSat

retrieval. Moreover, cloud top height in the model is significantly underestimated by about
1 km. However, below 6 km altitude, the modeled along-track averages of IWC are in ex-
cellent agreement with observations. The horizontal distribution of mixed phase clouds
(not shown) differs considerably, as significant amounts of IWC occur to the north of 51° N
in the observations, but not in the model. The comparison of the time series of IWC ob-

tained with Cloudnet for the grid cell of the TROPOS site, Leipzig shows quite similar
results (Fig. 8b). While IWC steadily increases above an altitude of 7 km and reaches up to
$4 \times 10^{-3}\,\mathrm{g\,m^{-3}}$ at 11 km altitude in the radar profile, the modeled profile shows IWC values
fluctuating around $10^{-4}\,\mathrm{g\,m^{-3}}$ in the same altitude range. However, below 6 km, modeled
IWC values are with about $1 \times 10^{-2}\,\mathrm{g\,m^{-3}}$ significantly higher than the observed values of

$1 \times 10^{-3}\,\mathrm{g\,m^{-3}}$. In summary, these results confirm the results from the infrared image com-
parison, as they give a quantitative estimate of the lack of IWC inside cirrus clouds. Cloud
infrared absorption and thus simulated cloud top temperatures are strongly dependent on
IWC until saturation begins above a value of $5 \times 10^{-2}\,\mathrm{g\,m^{-3}}$.

In addition, model precipitation amounts are evaluated, since they also may be affected by

the effects of mineral dust. Moreover, we can indirectly infer the distribution of precipi-
tating mixed phase clouds during the evaluation period, which is obscured in the satellite
images (Fig. 2). In Fig. 9a, measured 36 h-totals of precipitation (4 April, 00:00 UTC to 5
April, 12:00 UTC) at the various DWD stations in Germany and surrounding areas are com-
pared to the model data. The highest amounts of up to 30 mm rain occurred in northern

Germany associated with a nearly stationary warm front. Other than that, isolated precip-
itation occurred partly triggered by orography, as well as in a curved band over western
Germany with totals mostly below 10 mm. While the model agrees well in the maxima
of precipitation in association with the warm front, the location of the precipitation band
does not match the observations, as it is displaced too far to the northeast over the Baltic



Sea (54.5 °N, 11.0 °E). Even more remarkable is the occurrence of intense precipitation of locally more than 30 mm in the model linked with convection over western Germany on late 4 April, which was not observed at all. As a result, domain-wide precipitation is over-estimated in the SMBLK run by 64 %.

### 4.2    Dust sensitivity model runs

Having evaluated the model performance of the control run SMBLK in the previous Section, this Section is dedicated to the results of the sensitivity study. Again, we begin with the comparison of simulated infrared images to corresponding MSG satellite images and the estimation of upper tropospheric IWC, which is closely related to infrared temperatures. To answer the question whether the interactive dust simulation impacts cloud top

temperature in Fig. 10, we compare the run with climatological dust impact (Fig. 10a,c and e) to the run IINT with fully interactive dust-cloud and dust-radiation effects (Fig. 10b,d and f). The run IINT has more extensive cloud cover due to the impact of the simulated dust plume. In the central and southeastern parts of the domain, which are basically cloud-free in ICLM but cloud-covered in the MSG images (see Fig. 7), at least fragmented cloudiness

with cloud top temperatures around 220 K is present in IINT. However, the representation is still not very realistic, as a coherent cloud shield is still not present (see Fig. 7a,c,e for comparison). For a more quantitative estimation of the differences seen in the simulated images and the satellite images, a threshold temperature $T_{tr} = 240$ K is defined, to calculate the percentage of pixels colder than $T_{tr}$ in an hourly image series obtained over the period

from 4 April, 00:00 UTC to 5 April, 12:00 UTC. We are aware that with this approach the discrimination between cirrus clouds and other cloud types (e.g. convective clouds) cannot be assured and furthermore semi-transparent cirrus clouds with brightness temperatures $T_b > 240$ K are not considered. However, given the circumstances of this case study (i.e. the predominant occurrence of very cold widespread cirrus clouds in the satellite images),

this should give a reasonably good quantitative estimate of model performance regarding cirrus cloud representation. For the respective MSG image series, this value is 76 %, which highlights that for most of the time and area, the model domain D2 was covered by cirrus clouds. For the run ICLM, this value is 11 %, but increases twofold to 21 % with fully inter-active dust effects to in IINT. Nevertheless, this is still only 28 % of the observed value. In

order to trace this sensitivity seen in the infrared images back to cloud microphysics, we compare vertical profiles of IWC for the CloudSat overpass and the time series at the TRO-POS site. Figure 11 shows the vertical profiles of the model-to-measurement ratio for the different runs. The CloudSat comparison reveals an expectedly strong sensitivity to mineral dust, as IWC increases between 200 % and 500 % in the run IINT compared to ICLM.

In the altitude range between 7 km and 11 km, about 10 % of the observed IWC is present





in the run ICLM, but this value increases to about 40 % in the run IINT. The gap to observations is however larger in the model near the tropopause at 11.5 km altitude where it is in the order of one magnitude (Fig. 11a). At the TROPOS site, a large increase in IWC can be seen in the upper troposphere, when considering interactive dust effects in the model

(Fig. 11b). Above 9 km altitude, only about 5 % of observed IWC is present in the run ICLM with climatological mean dust effect, but this value increases again to roughly 40 % above 9 km altitude in the run IINT with fully interactive dust effects. Moreover, in IINT, also the representation of precipitation is improved in comparison to the run with single-moment bulk microphysics (see Fig. 9b). The geographical distribution of the precipitation band

over northern Germany matches observations better in IINT than in SMBLK. The intense precipitation band seen in the run SMBLK over western Germany is still present in IINT, but shows significantly lower totals of up to 20 mm. Generally, precipitation is better represented by IINT than SMBLK, as domain-wide precipitation is overestimated in by only 6 % in IINT but 64 % in SMBLK. The correlation coefficient between observational and model

data can be used to estimate the agreement in the geographical distribution of precipitation. The correlation coefficient is 0.5 in IINT, but only 0.4 in SMBLK, suggesting a better representation of precipitation in the model run with interactive dust.

To disentangle the different contributing dust-effects to the improved cloud representation, we evaluate the additional sensitivity model runs RCLM and CCLM, which have climato-

logical dust-radiation and dust-cloud-activation effects, respectively. Both runs do not differ significantly from the run IINT in the infrared simulation, suggesting heterogeneous ice nucleation of desert dust being the main contributor to the development of the additional cloudiness. This is also reflected in the CloudSat comparison, as even there, the results of RCLM and CCLM do not show a clear shift towards higher or lower IWC values compared

to IINT. Only in the comparison at the TROPOS site, there is 80 % less IWC modeled in the mixed phase cloud region below 6 km altitude without radiative effects of the dust plume in comparison to IINT. Supposedly, dust-radiative effects can affect mixed phase clouds in multiple ways and not only by reducing cloud cover through stabilization of the thermal stratification of the atmosphere. Dust-radiation effects are also evident in the precipitation

analysis. Comparing the results with fully interactive dust effects (Fig. 9b) to the results of climatological dust-radiation effects (Fig. 9c), more widespread precipitation is evident over southern Germany in latter case, which is reflected also in a stronger overestimation of precipitation by 13 % in RCLM versus 6 % in IINT. This improvement, however, does not increase the correlation with observations, with a correlation coefficient of 0.5 for IINT and

RCLM respectively.

The last sensitivity model run IAIP is used to evaluate the alternative INP-parameterization P08 in comparison with the parameterization U17. This is to assess a range of uncertainty





due to uncertainties in the freezing properties of mineral dust. In the infrared satellite images, IAIP produces the lowest cirrus cloud cover with only 7 %, which is 3 times lower compared than in IINT. Similarly, in the CloudSat comparison in Fig. 11, IAIP shows the lowest IWC values, except near the tropopause, where the run SMBLK shows the lowest values. Above 6 km altitude, the difference between IAIP and IINT is particularly large with mostly only 10 % IWC present in IAIP. Even more drastic differences are seen in the comparison at TROPOS at 9 km altitude, where only 0.1 % of the observed IWC are present in the IAIP run. For the tested INP-parameterizations, INAS-density as the measure of activity differs by the order of several magnitudes in the deposition freezing range (not shown here). These uncertainties obviously affect the model performance to a similar extent than the uncertainties in the representation of mineral dust concentrations in the model.

Table 3 summarizes the most important findings of the sensitivity studies presented in this Section.

**Table 3.** Summary table of the sensitivity study: the cirrus cloud fraction estimation is based on the infrared temperature simulation. IWC in cirrus clouds is the calculated mean IWC inside the layer above 9 km altitude for both comparisons (CloudSat and TROPOS). Precipitation deviation is calculated as the relative deviation of the sum of modeled 36 h-totals at all stations from the sum of corresponding measurements.

| Data | Cirrus | | Precipitation | |
|---|---|---|---|---|
| | cloud cover [%] | IWC [$\mathrm{g\,m^{-3}}$] | overestimation | correlation |
| **Observation** | 76 | $5 \times 10^{-3}$ | - | - |
| **SMBLK** | - | $9 \times 10^{-4}$ | 59 | 0.38 |
| **ICLM** | 11 | $7 \times 10^{-4}$ | 21 | 0.50 |
| **IINT** | 21 | $2 \times 10^{-3}$ | 6 | 0.50 |
| **RCLM** | 22 | $2 \times 10^{-3}$ | 13 | 0.53 |
| **CCLM** | 21 | $2 \times 10^{-3}$ | 8 | 0.49 |
| **IAIP** | 7 | $3 \times 10^{-4}$ | -4 | 0.34 |

### 4.3 Detailed evaluation of cloud microphysics

In this section, detailed cloud-microphysics data of the ML-CIRRUS flights conducted on 3 April and 4 April, respectively, are analyzed. A description of the data sets used and the methodology for data processing are presented in Sect. 2.3.4. In a second part, the ML-CIRRUS data from 3 April is compared to model results, and the sensitivity of modeled ice



particle diameter $D_i$ to heterogeneous ice nucleation of mineral dust is evaluated.

Observed IWC along trajectories of both flights on 3 April and 4 April is shown in Fig. 12 underlaid with MSG infrared of cloud cover. Accordingly, on 3 April, HALO did not

sample the main cloud shield but broken cirrus clouds at 8 to 12 km altitude. In addition, a stratiform cloud shield below 7 km altitude was probed during ascent and descent at start and landing. IWC within cirrus clouds reached up to $10^{-2}\,\mathrm{g\,m^{-3}}$ during this flight (Fig. 12a). On 4 April, HALO headed westward to Portugal and passed at an altitude of 9.5 km through a dense band of cirrus clouds located to the north of the Alps (Fig. 12b). Cloud

top temperatures of this cirrus were much lower than those of cirrus clouds sampled on 3 April. IWC measured within the clouds on 4 April is partly above $10^{-1}\,\mathrm{g\,m^{-3}}$, which is more than one order of magnitude higher than on 3 April.

Figure 13 shows ice particle number densities $[\mathrm{m^{-3}\,\mu m^{-1}}]$ as colored shadings for both flights averaged along the flight trajectory. Accordingly, on 3 April, the highest particle

densities can be found in the smallest size bins with particle diameters lower than $67.5\,\mu m$ (Fig. 13a). Especially high particle number densities of up to $5 \times 10^3\,\mathrm{m^{-3}\,\mu m^{-1}}$ occur at the cloud tops, indicating the influence of aviation induced contrail cirrus. The decline in particle number density with increasing diameter is monotonic, with the largest particles having a diameter of around $200\,\mu m$. The number-averaged mean diameter $\overline{D}_N$ depicted

by the solid line and the top axis in Fig. 13 increases nearly linearly from $25\,\mu m$ at 11 km altitude to $55\,\mu m$ at 9 km altitude. This suggests diffusional growth of settling ice particles before they begin to sublimate below 9 km altitude near the cloud base. According to measured prevalent particle sizes, cirrus clouds sampled on 3 April can be classified as in situ origin cirrus (Luebke et al., 2016). For the cloud sampled on 4 April, the main distinguish-

ing characteristics of the clouds sampled on 3 April are the much broader PSD along with much higher particle number densities. This is especially pronounced near $20\,\mu m$ with values up to $10^5\,\mathrm{m^{-3}\,\mu m^{-1}}$, which is close to 100 fold the value measured on 3 April. The PSD again shows a monotonic decrease of particle densities with increased particle sizes, with no hints of a bi-modular structure. $\overline{D}_N$ is between $35\,\mu m$ and $40\,\mu m$ in the height range of

8 to 9.5 km, which is slightly lower than $\overline{D}_N$ on 3 April.

On basis of these observations, the optically dense cirrus cloud sampled on 4 April can also be classified as in situ origin cirrus, as small particles are obviously dominating. Furthermore it is likely, that the additionally available humidity to be converted into IWC on 4 April was not consumed by larger ice crystals but new ice particles. This points toward

a higher INP concentration on 4 April than on 3 April, which is supported by NIXE-CAS aerosol measurements in the diameter range of $0.6 - 2\,\mu m$.

Moreover, simulated dust-particle number concentrations $n_d$ inside sampled clouds for both flights differed significantly (not shown), as on 3 April $n_d$ ranged from $3 \times 10^4\,\mathrm{m^{-3}}$





to $2 \times 10^6\,\text{m}^{-3}$ but on 4 April $n_d$ was consistently above $4 \times 10^6\,\text{m}^{-3}$ and reached even up

to above $10^7\,\text{m}^{-3}$ for a small part of the flight track. This underpins the assumption of the higher INP-concentrations affecting clouds on 4 April.

To assess whether the model shows a similar behaviour with enhanced heterogeneous ice nucleation, scatter plots of modeled $D_i$ for the three sensitivity model runs ICLM, IINT and IAIP are plotted in Fig. 14 against the ML-CIRRUS data for 3 April. Furthermore,

values of $\overline{D}_N$, as well as $\overline{D}_{IWC}$ are calculated for all data sets (see Eq. 13 and 14). In the run ICLM, which considers the climatological dust concentration, data points are shifted to too high values of modeled vs. measured $D_i$ for high IWC values larger than $10^{-3}\,\text{g\,m}^{-3}$, while the contrary is the case for the IWC values lower than $10^{-3}\,\text{g\,m}^{-3}$ (Fig. 14 a). This asymmetry results in $\overline{D}_N = 27\,\mu\text{m}$ for the model, which is smaller than the measured value

of $38\,\mu\text{m}$. The contrary is the case for the IWC-weighted average, which puts more weight on the larger ice particles, with $\overline{D}_{IWC} = 72\,\mu\text{m}$ in the model and $\overline{D}_{IWC} = 53\,\mu\text{m}$ in the measurements. This suggests, that at low values of IWC, INPs activate very efficiently in the model with the parameterization U17. However, at larger IWC values, supposedly there are not enough INPs available (constrained by the climatological mean dust concentration)

in ICLM for the formation of additional ice crystals. With interactive dust in IINT, we observe a large decrease in $D_i$ in the denser cloud parts with high IWC (Fig. 14 b). For most of the data points, simulated $n_d$ is above the climatological mean, thus making additional INPs available for ice nucleation in the parts of clouds, while ice nucleation is limited in ICLM. This is reflected in a 46% decrease of $\overline{D}_{IWC}$ to $39\,\mu\text{m}$, while $\overline{D}_N$ decreases by only

8% to $25\,\mu\text{m}$ compared to ICLM. With the application of the parameterization P08 in the run IAIP, for the whole IWC spectrum, modeled $D_i$ is larger than for the measurements, suggesting the presence of abundant potential INPs, which, however, do not activate efficiently (Fig. 14 c).

Figures 14d to 14f show simulated infrared images for the runs ICLM, IINT, and IAIP, re-

spectively. From the sensitivity of cold cloud cover to ice nucleation we can conclude that the decrease in $D_i$ in dense cloud regions in response to raised INP-concentrations (positive Twomey effect) is the main reason for the additional cirrus cloud formation in the model, as it occurs significantly only in the run IINT. A larger number of ice crystals increases the cloud optical depth (COD), which our model can account for, as the cloud-radiation scheme

uses prognostic effective diameters $D_i^{ef}$ from the two-moment microphysics scheme (see Sect. 2.1.4). This increase in COD increases long-wave emissions at the cloud tops, further cooling these regions and enhancing updraughts inside cirro-cumulus clouds. This eventually causes cloud cover to increase in our model. However, cloudy parts which are less dense are not susceptible to this feedback mechanism, as otherwise we would observe an

increase in cloud cover using U17 independently of the representation of mineral dust in



the model. Thus, higher than climatological mean mineral dust concentrations in the model are necessary for additional cirrus cloud formation in this study. Our reasoning is further underpinned by the fact, that without detailed cloud-radiation interactions, no significant sensitivity to mineral dust can be observed in our model runs.

In order to assess the limitations in cirrus cloud representation in the simulations due to an inaccurate representation of the thermodynamic environment, we further evaluate vertical profiles of specific humidity (SH) obtained from radiosoundings over Germany (see Sect. 2.3.5). Figure 15a shows radiosonde profiles for the dates 4 April, 00:00 UTC, 4 April, 12:00 UTC and 5 April, 00:00 UTC, respectively. There is a tendency of water vapor to in-

crease over the considered time period in a layer at approximately 7 km altitude, which is most pronounced between 4 April, 00:00 UTC and 4 April, 12:00 UTC, when the dense cirrus canopy appears in the simulation domain. Figure 15b shows modeled to measured ratios of SH for the IINT run. On 4 April, 00:00 UTC, the model has 40 % higher values of SH below 8 km than measured. Above 11 km altitude, the modeled SH is significantly

higher than in the observed, but this seems more likely caused by the inaccurate representation of tropopause height in the model. On 4 April, 12:00 UTC, there is less SH present in the model in an altitude range between 7 km and 11 km. This becomes even more evident on 5 April, 00:00 UTC, when only 60 % of measured SH is present in the model. Thus, the model does not seem to follow the trend of increasing SH over time as seen in the ra-

diosonde profiles. There can be different reasons for these observations. Obviously, there is a large vertical gradient in SH, and turbulent transport associated with cloud development could easily modify the vertical distribution of SH. However, the layer with the relatively dry air is located at the cirrus cloud base or below. So turbulent mixing associated with cirrus formation should not affect this layer at all, and in fact, no significant differences are

found between the model runs. More likely this is caused by an inaccurate representation of humidity in the meteorological boundary data. Supposedly, the thermodynamic environment inside dust plumes is affected by the dust in manifold ways during the long-range transport from the Sahara to Europe, which was not represented in the boundary data.

**5 Summary and Conclusions**

The aim of this study was to model the impact of desert dust on cloud development for the dust outbreak in early April 2014. Therefore we used the regional dust model COSMO-MUSCAT to simulate dust emission and dust transport from northern Africa toward Europe, as well as dust-radiation and dust-cloud interactions over Germany. For reference,

a first model run with COSMO using the operational single-moment bulk microphysics





scheme and without dust-radiation and dust-cloud interactions was performed for Germany on a grid with 2.8 km horizontal spacing over the period 3 April, 12:00 UTC to 5 April, 12:00 UTC. Simulated cloud top temperatures of this model run were compared to MSG infrared imagery, and modeled IWC to cloud radar retrievals from CloudSat satellite

and measured at TROPOS site in Leipzig. We qualitatively found a large deficit in modeled cirrus cloud cover over the 48 h period, which coincided with above-average mineral dust concentrations throughout the height range of the troposphere. The lack in cirrus clouds was also seen in the comparison of IWC, as only 10 % of observed IWC was present in the model at the relevant altitudes. Moreover, modeled precipitation were overestimated

by 64 % on average if compared to DWD station records. Thus, these results showed a strong misrepresentation of cloudiness and precipitation, possibly linked to the neglect of interactive dust effects in the model during the Saharan dust event. By interactive dust modeling with a two-moment microphysics scheme (Seifert and Beheng, 2006) including dust-radiation, dust-cloud activation and detailed dust-ice nucleation effects, the question

is investigated whether with this more detailed approach cloud and precipitation representation can be improved in the model. Based on the Saharan dust distribution modeled with COSMO-MUSCAT on a domain with 14 km grid spacing and covering northern Africa and Europe, 5 sensitivity model runs were performed with COSMO-MUSCAT with 2.8 km horizontal resolution over Germany. The setup of these runs included a run with fully interac-

tive dust effects, a run with a spatially and temporally uniform cimatological low average dust concentration prescribed in the radiation and cloud schemes and two runs with climatological dust-radiation effects and dust-cloud activation effects, respectively. While for these model runs the ice parameterization of Ullrich et al. (2017) (U17) was used, an additional fully interactive model run using the parameterization of Phillips et al. (2008) (P08)

was performed to assess the uncertainty related to the choice of the ice parameterization. The evaluation of the modeled dust fields with AERONET data and an aerosol extinction profile retrieved from lidar data at the TROPOS site in Leipzig showed a reasonably good agreement with observations. Uncertainties can be related to the presence of other aerosol components (e.g. black carbon and organic aerosol), which were not simulated

with COSMO-MUSCAT.

Comparing the two model runs either using fully interactive or fully prescribed from climatology dust interactions, we found a strong sensitivity of cloud formation to mineral dust concentrations. Cirrus cloud cover doubled and IWC increased by a factor of 2 to 8 inside the cirrus layer, reaching up to about 40 % of the observed values from CloudSat

satellite and TROPOS site cloud radar, due to the feedback of the online-simulated dust concentrations.

This sensitivity was found primarily due to increased heterogeneous ice nucleation of min-

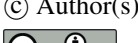



eral dust, as the two additional sensitivity model runs using climatological mean mineral dust concentrations either for radiation interaction or cloud activation respectively, pro-
duced the same results for cirrus clouds as with the fully interactive run. Radiative effects of the dust plume predominantly had an impact on precipitation formation in southern Germany. Dust-cloud-activation effects were not discernible, which suggests that in our study the assumed aerosol background dominated this process even in the presence of the simulated mineral dust concentrations.

Lastly, the choice of the INP-parameterization turned out to be at least as important as the application of a more realistic spatially and temporally varying mineral dust distribution, and the parameterization of Ullrich et al. (2017) led to the best agreement of model results with observations.

Evaluating ice cloud microphysics in more detail with in situ measurements obtained dur-
ing the ML-CIRRUS aircraft campaign showed that in general the two-moment microphysics scheme of Seifert and Beheng (2006) along with the modifications done to include detailed dust-cloud interactions was capable of reproducing realistic particle diameters inside cirrus clouds. Homogeneous ice nucleation of liquid aerosol particles was not important in our model, as supersaturation over ice was consistently below the homogeneous
freezing threshold given in Kärcher and Lohmann (2002). Thus, heterogeneous ice nucleation controlled particle concentrations and diameters inside in situ formed cirrus clouds. In fact, we could see a strong sensitivity of modeled ice particle sizes to mineral dust concentrations, and the smallest particle diameters with best agreement to measurements were found using interactive dust simulation with the U17 parameterization. The occurrence of
the positive Twomey effect in response to high mineral dust concentrations fostered the development of more extensive cirrus clouds in our model due to dust-cloud-radiation effects represented in the radiation scheme extended by Dipu et al. (2017).

The improvements seen in our model still cannot be considered adequate for a realistic representation of cirrus cloud cover. This limitation was likely inflicted by the meteorological
boundary fields, which underestimated humidity in layers relevant for cirrus cloud formation. In this regard, it would be useful to raise specific humidity inside selected vertical layers in the boundary data to match observed values in order to test the cloud representation and sensitivity to mineral dust in additional model runs with adjusted input data. However, this was beyond the scope of this paper.

Based on the outcomes of the present sensitivity study, we recommend to test this or similar modeling approaches for other individual cases, which are characterized by extensive cirrus cloud development in association with major desert dust outbreaks into the midlatitudes. It has to be shown, on a statistical basis, whether weather forecast quality during such periods can indeed take advantage of the more detailed but also numerically more




expensive interactive weather-dust simulations. For future research, more field studies investigating microphysical properties of cirrus clouds affected by mineral dust could provide valuable information for model evaluation to corroborate our findings. Multi-spectral satellite observations and derived cloud products can further improve the ability to characterize the spatial distribution of cirrus clouds with lower optical thicknesses to compare

their representation in simulations. In addition, further aircraft in situ ice nucleation experiments are needed to reduce the remaining uncertainties in parameterized INP properties of mineral dust and aerosol in general.

**Acknowledgments**

The authors thank the Deutscher Wetterdienst (DWD) for good cooperation and support.

We are grateful for computing time from the German Climate Computing Center (DKRZ). The research leading to these results has partly received funding from ACTRIS-2 in HORIZON 2020 under grant agreement no. 654109. CV thanks for funding by the DFG within the SPP1294 HALO under contract no VO 1504/4-1 and by the Helmholtz Association under contract no W2/W3-60. We thank the PI investigators and their staff for establishing and

maintaining the AERONET sites used in this investigation. EUMETSAT is acknowledged for providing MSG SEVIRI data. CloudSat data has been provided by the NASA Langley Research Center Atmospheric Science Data Center, and Atmospheric Sounding profiles by the University of Wyoming.

**Appendix A**

**Infrared satellite imagery simulation**

To take advantage of prognostic cloud particle number concentrations in the runs using two-moment microphysics for synthetic infrared image derivation, an appropriate radiative transfer model was constructed in the framework of this study. It is based on absorption and emission of black body radiation at thermodynamic equilibrium and neglects

scattering of thermal radiation. The absorption coefficient is calculated as the linear combination of the individual absorption by the modeled six hydrometeor classes, as well as absorption by water vapour. For cloud ice and cloud water, additionally, parameterized sub-grid scale mass contents are considered. The selected PSD for all classes is a generalized gamma function, in accordance with the two-moment microphysics scheme, and

parameters are adopted from the currently used version in this study.
For the spherical particle classes graupel, rain and hail, which contain exclusively particles larger by orders of magnitude than the considered wavelength, a constant absorption ef-



ficiency of 0.95 is assumed for calculating absorption coefficients based on the geometric cross section. Cloud ice and snow are treated in common, and are assumed to consist of

hexagonal plates. For this particle class, the absorption coefficient is parameterized using polynomial approximations to exact Mie-calculations (Fu et al., 1998). Therefore, $D_i^{ef}$ is inferred from $D_i$ by taking the size-dependent linear dimension relationships for hexagonal plates given in Fu et al. (1998). Cloud droplets are treated as spherical particles and the absorption parameterization of Lindner and Li (2000) is used. To account for water vapor

absorption an empirical formulation based on the water vapor mixing ratio being valid for a reference pressure and temperature is scaled to the actual pressure and temperature value (Chou and Suarez, 1994). The emitted radiation of the Earth's surface is approximated with a surface emissivity ranging from 0.95 for surface temperatures lower than 273 K to 0.85 for surface temperatures above 280 K. To account for the effects of a slanted satellite viewing

angle, the derived optical thicknesses are inversely scaled with the cosine of the satellite viewing angle, calculated for the satellite position 0 °N, 0 °E and the geographical position of the referred model grid point.

To make the infrared simulation also applicable to model data with single-moment microphyisics, particle diameters either have to be assumed or parameterized of other prognos-

tic quantities. Here, we assume a constant diameter of $D_c = 10\,\mu m$ for cloud droplets and $D_r = 10^3\,\mu m$ for rain droplets. $D_i$ is parameterized with IWC, containing the contributions of modeled $q_i$ and $q_s$. A parameterization was constructed, using ML-CIRRUS data for 3 and 4 April respectively, and is given by the following analytical expression:

$$D_i = \left[0.044531 \times \log_{10}(\text{IWC}) + 1.554498\right]^{0.1}, \tag{A1}$$

where IWC is given in units of $\left[\text{g m}^{-3}\right]$ and $D_i$ in $[\mu m]$. This expression gives reasonable $D_i$ ranging from 8 μm for $\text{IWC} = 10^{-7}\,\text{g m}^{-3}$ to 65 μm for $\text{IWC} = 10^{-1}\,\text{g m}^{-3}$.

For the single-moment bulk microphysics run, and using this parameterization, we compared simulated brightness temperatures with the more comprehensive (treatment of scat-

tering) Radiative Transfer model for TIROS Operational Vertical sounder (RTTOV) model (Saunders et al., 2018), and found a very good agreement with our radiative transfer scheme, as simulated temperatures differed by not more than 5 K, which is in the order of typical uncertainties (Senf and Deneke, 2017).





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





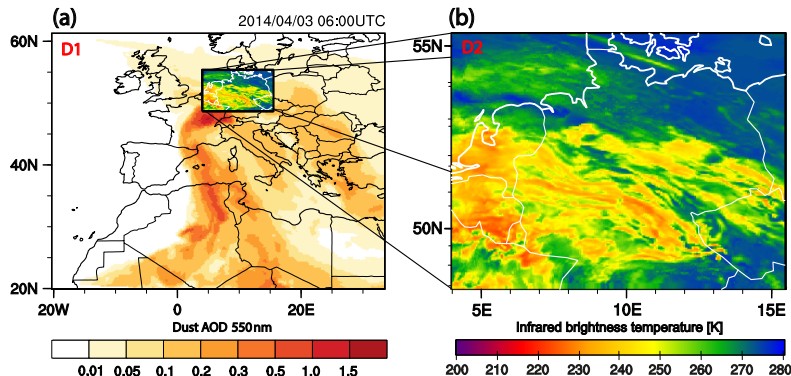

**Fig. 1.** (a) Model simulation domain D1 with 14 km grid spacing showing dust AOD fields by COSMO-MUSCAT. (b) Inner model domain D2 with 2.8 km grid spacing showing simulated infrared brightness temperatures of a COSMO-MUSCAT run with interactive dust effects on cloud microphysics.

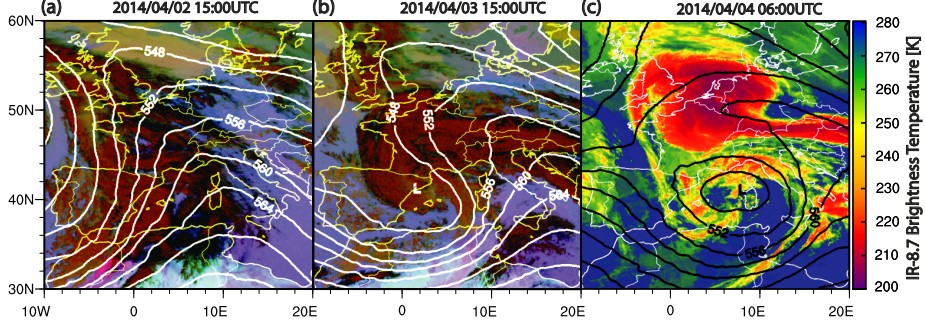

**Fig. 2.** (a, b) Meteosat Second Generation (MSG) SEVIRI dust composite images for 2 April, 15:00 UTC and 3 April, 15:00 UTC, respectively. (c) MSG SEVIRI IR-8.7 brightness temperature for 4 April, 06:00 UTC. All images are overlaid with 500 hPa geopotential height contour lines from COSMO-MUSCAT (14 km) with 4 dam spacing.



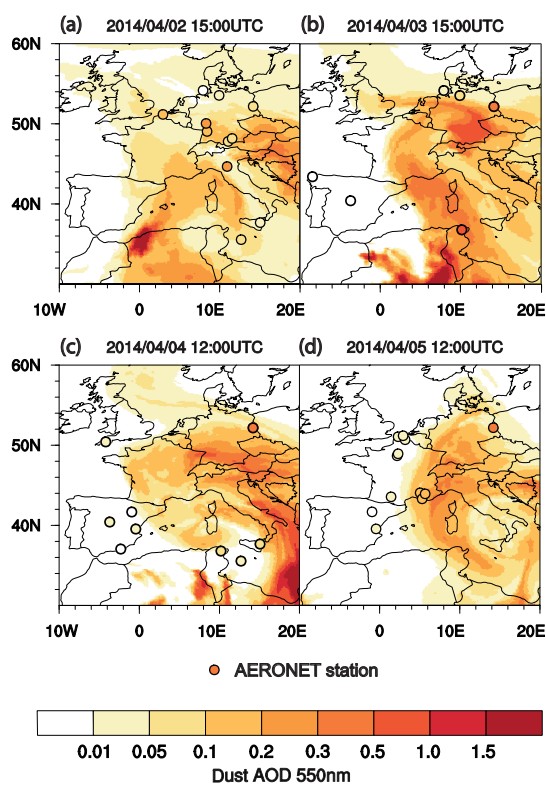

**Fig. 3.** Maps of dust AOD at 550 nm as simulated with the dust transport model COSMO-MUSCAT on the D1 grid for the dates 2 April, 15:00 UTC, 3 April, 15:00 UTC, 4 April 12:00 UTC, and 5 April, 12:00 UTC, respectively. AERONET observations of coarse mode AOD at 500 nm are marked by coloured circles.



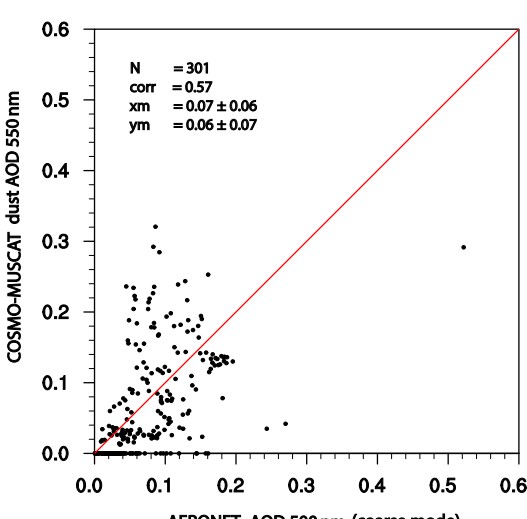

**Fig. 4.** Scattter plot of observed 500 nm coarse-mode AOD and modeled dust AOD from the 14 km COSMO-MUSCAT run for all available AERONET stations within domain D1 (see Fig. 1). The data pool was gathered over the period from 2 April, 00:00 UTC to 6 April, 00:00 UTC by taking hourly model outputs and AERONET observations available within ±0.5 h of corresponding time frames into account. Parameters printed are number of data points in each set (N), correlation coefficient between both data sets (corr), mean of the observations (xm), and mean of modeled values (ym).





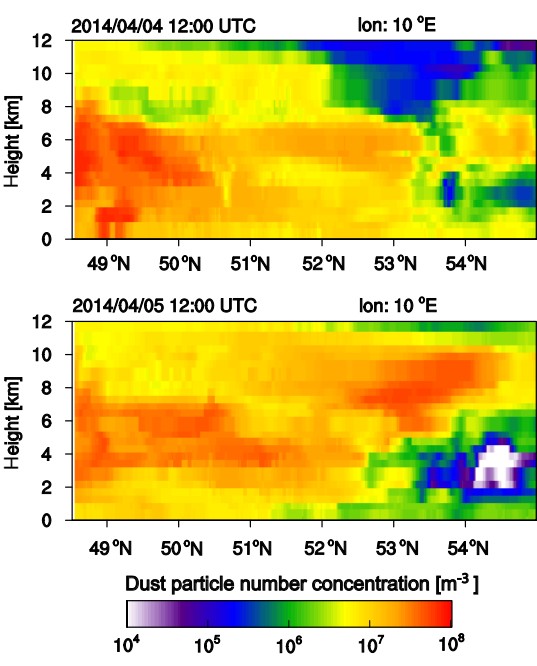

**Fig. 5.** Vertical cross section of dust particle number concentration along the meridian $10°$ E over Germany as computed with COSMO-MUSCAT at 2.8 km horizontal resolution for 4 and 5 April 2014 at 12:00 UTC respectively.



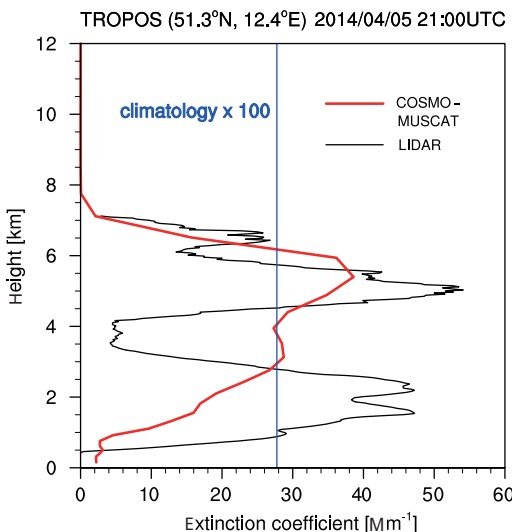

**Fig. 6.** Vertical profile of aerosol extinction coefficient retrieved from lidar observations obtained at TROPOS in Leipzig on 5 April, 21:00 UTC, and calculated from the modeled dust fields at the nearest grid point.

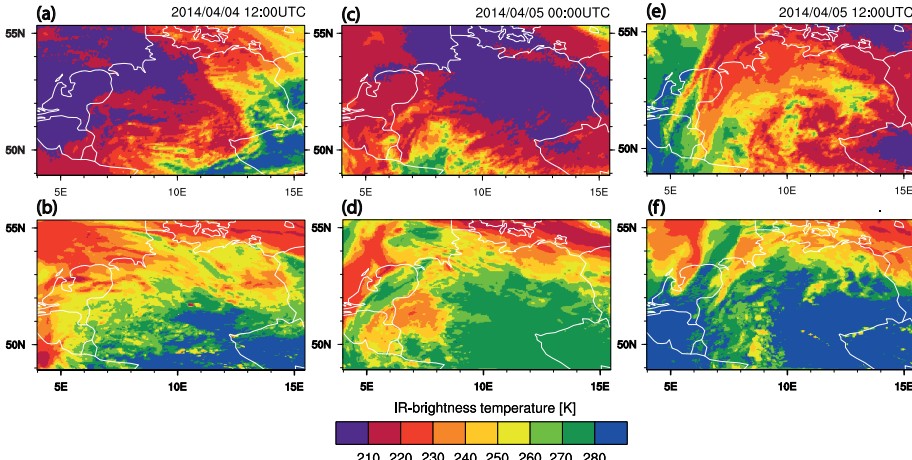

**Fig. 7.** Infrared (IR; 8.7 µm channel) brightness temperature maps based on (top row) MSG SEVIRI data of EUMETSAT (top row) and (bottom row) on the infrared simulation with the model output fields of the run SMBLK. Areas of brightness temperatures below 240 K are colored.





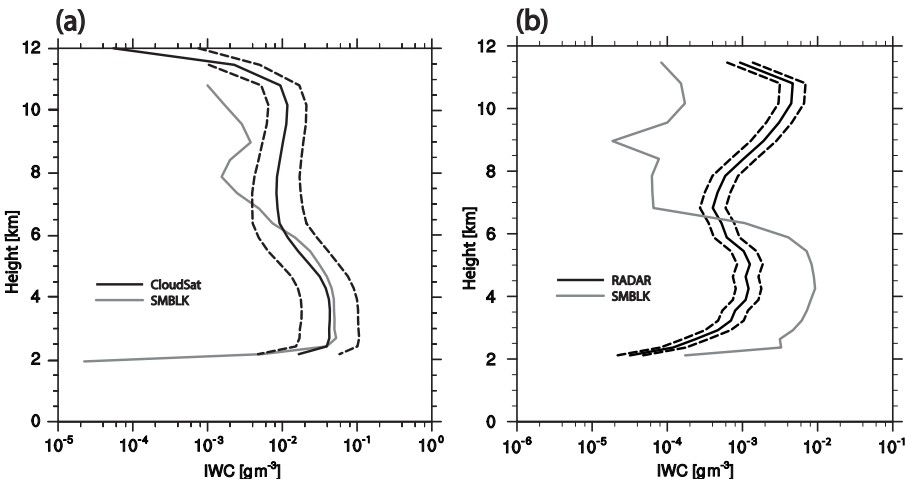

**Fig. 8.** (a) Spatially averaged vertical profiles of IWC retrieved from cloud radar and the model run SMBLK along the CloudSat satellite overpass on 4 April, 12:30 UTC. (b) Similar to (a), but for the time averaged radar observations and model data at the TROPOS site in Leipzig (51.3° N, 12.3° E). Absolute values are depicted by full lines, while dashed lines mark the averaged measurement uncertainty of cloud radar retrievals.



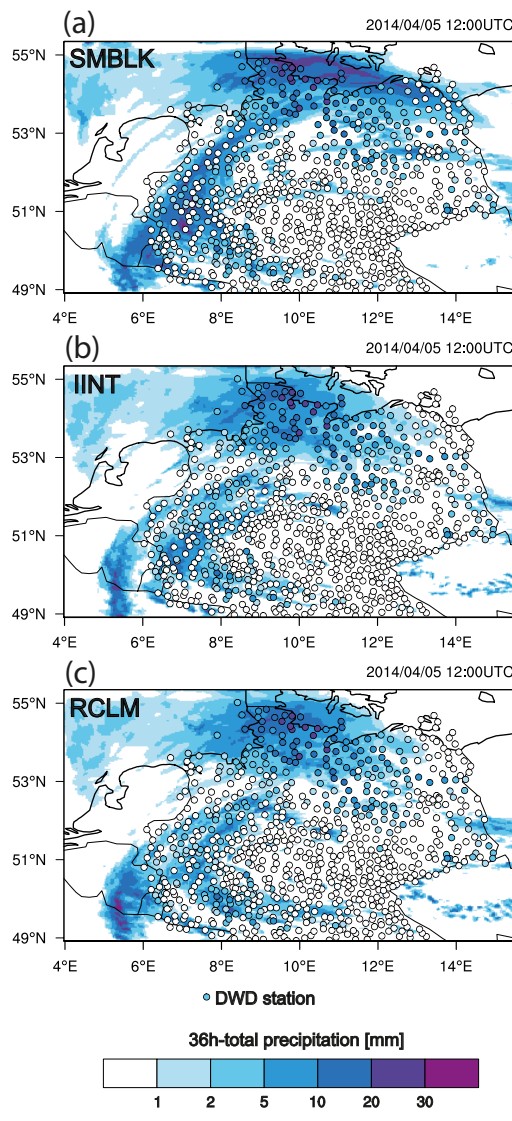

**Fig. 9.** Maps of modeled 36 h-precipitation totals for the different sensitivity model runs (a) SMBLK, (b) IINT, and (c) RDLM overlaid with station precipitation measurements (black rendered, color-coded circles) from DWD stations.





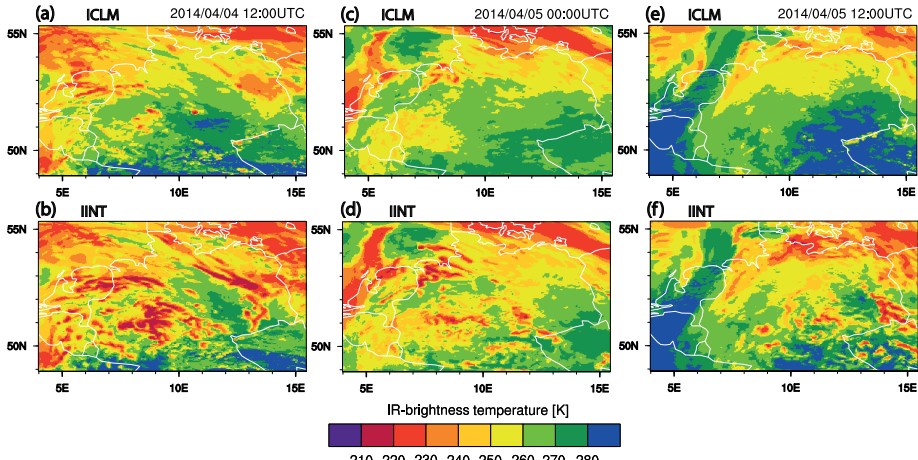

**Fig. 10.** Simulated infrared images of brightness temperature as in Fig. 7 for (top row) the run with climatological mean dust interactions (ICLM) and (bottom row) interactive dust effects (IINT). Areas of brightness temperatures below 240 K are colored.

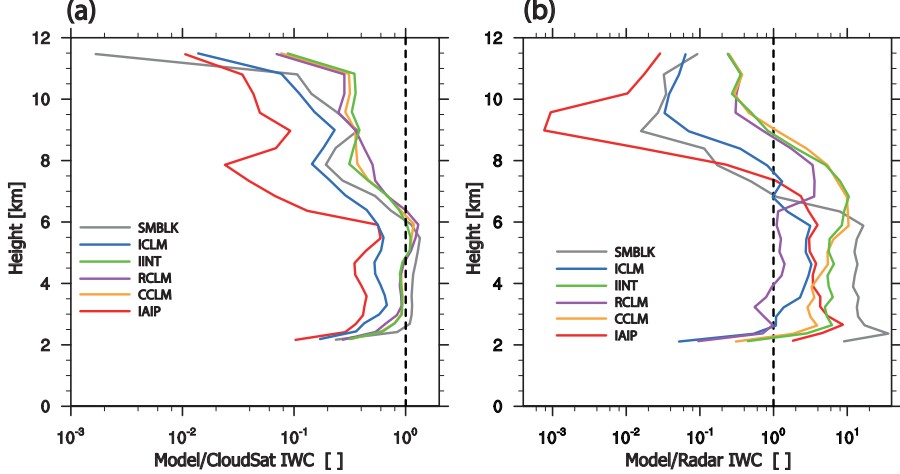

**Fig. 11.** Vertical profiles of model-measurement ratios of IWC for (a) the CloudSat overpass on 4 April, 12:00 UTC and (b) for the time averaged radar observations at TROPOS site in Leipzig (51.3 ° N, 12.3 ° E).




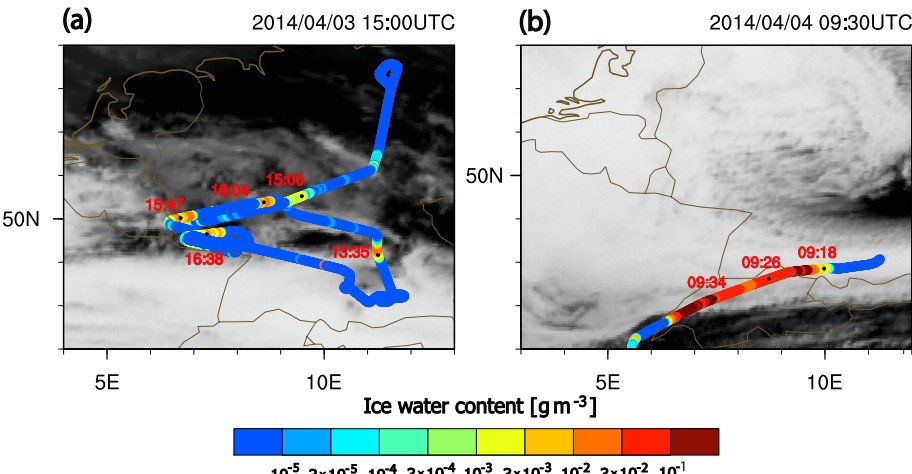

**Fig. 12.** Trajectories of ML-CIRRUS flights conducted (a) on 3 April and (b) on 4 April respectively. The colors of the trajectories represent IWC above 7 km altitude, which is inferred from the measured ice PSD. In addition, flight time stamps in UTC are shown for both tracks. The background shows MSG SEVIRI IR-8.7 images from (a) on 3 April, 15:00 UTC and (b) on 4 April, 09:00 UTC respectively, indicating differences in cloud cover with cirrus on these days.

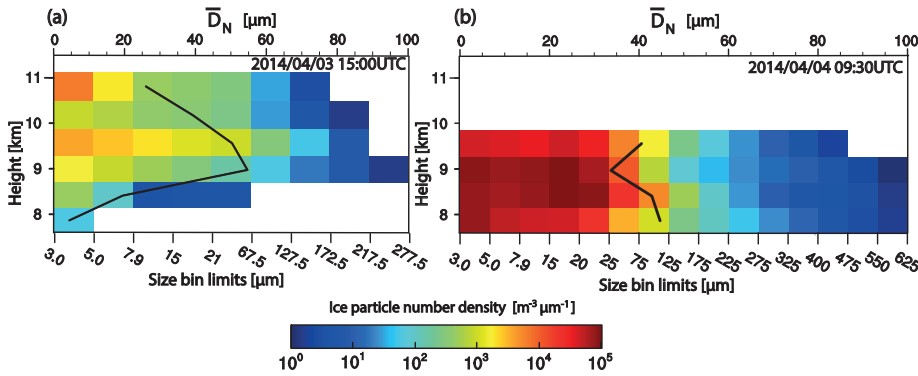

**Fig. 13.** Ice PSD from measurements aboard HALO aircraft during ML-CIRRUS campaign on (a) 3 April 15:00 UTC and (b) on 4 April 09:30 UTC. The area plots show vertical profiles of measured particle number density $\mathrm{d}n_i/\mathrm{d}d$ for the size bins (limits marked by the ticks of bottom x-axis) and interpolated to the 2.8 km COSMO-MUSCAT vertical levels. In addition for each layer, the number weighted volume mean particle diameter $\overline{D}_N$ is shown for measurements (top x-axis).



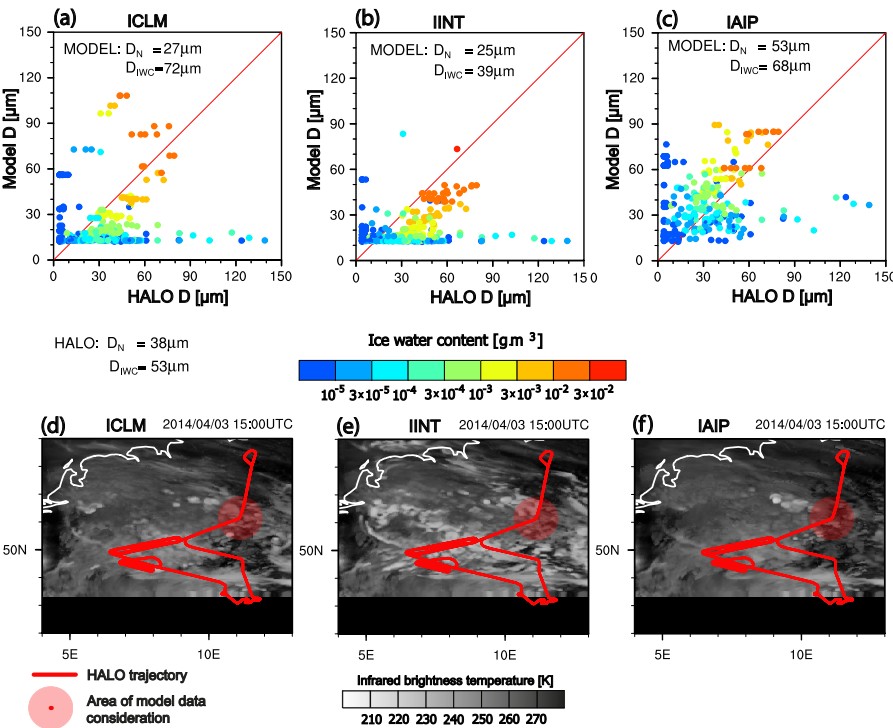

**Fig. 14.** (a, b, c) Scatter plots of ice particle diameter $D_i$ measured aboard HALO during ML-CIRRUS against model results for the model runs ICLM, IINT and IAIP, respectively. $\overline{D}_N$ is the number-weighted average of $D_i$, $\overline{D}_{IWC}$ is the IWC-weighted average of $D_i$. (d, e, f) Simulated infrared images for model runs ICLM, IINT and IAIP overlaid with HALO flight track and a reddish shaded circle, within which model data are considered for the comparison.





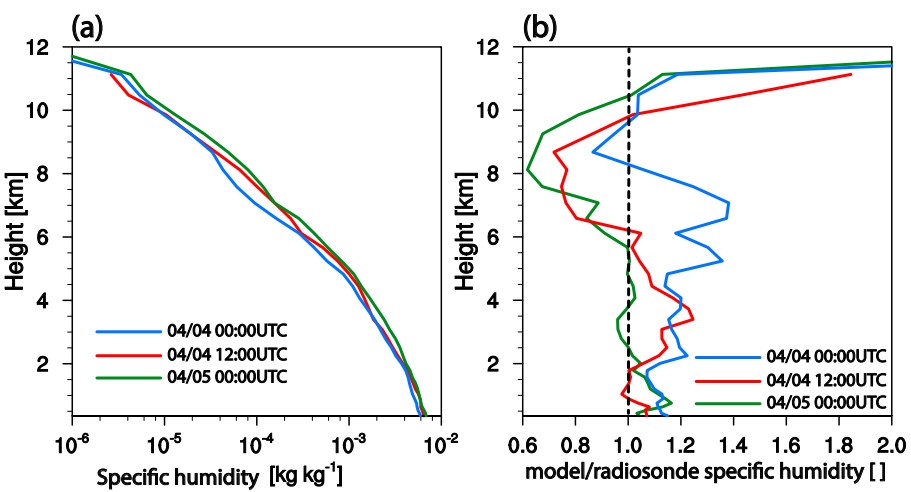

**Fig. 15.** (a) Averaged vertical profiles of specific humidity (SH) obtained from atmospheric sounding at all stations located within the domain D2 for the dates 4 April, 00:00 UTC, 4 April, 12:00 UTC, and 5 April, 00:00 UTC, respectively. (b) Averaged model-measurement ratios of SH as in (a).