# Peer review of "The impact of mineral dust on cloud formation during the Saharan dust event in April 2014 over Europe"

_Atmospheric Chemistry and Physics, 2018_

## Referee Comment (RC1) · Anonymous Referee #1 · 22 Aug 2018

The paper discusses the sensitivity of an upper-level cloud cover to two different microphysics schemes, with and without dust-cloud and dust-radiation feedbacks during a dust outbreak over Europe. It presents a comprehensive comparison between simulations and remote sensing observations of aerosol and cloud properties as well as in situ measurements from an aircraft campaign. The simulation with the dust-cloud and dust-radiation feedbacks provides the best results. This is attributed to enhanced deposition freezing. Different empirical ice nucleation parameterizations are then tested, which shows the importance of remaining uncertainties in the ice nucleating properties of mineral dust. Last, the best simulation is shown to be too dry in the upper troposphere, which is likely the main cause of underestimating the cloud cover. My suggestion at

the end of the reading would have been to redo the work with more realistic values of specific humidity at the initial and boundary conditions. This would however be too much work and as stated in the text, it is beyond the scope of the paper. Despite this disappointing result on cloud cover prediction, the paper presents an in-depth discussion on the impact of dust on cloud cover. As such the paper deserves publication to ACP.

**Minor comments**

Page 3, line 56. Extra "is" between "homogeneous" and "nucleation"

Page 4, line 104. Typo on "microphyiscs"

Page 19, line 587. The lidar measurements show large values of extinction coefficient around 2-km altitude. You implicitly attribute this signal to black carbon aerosol that can absorb visible radiation. This radiative effect is not present in the simulation. Because it is a strong signal, it might have a big impact on the stability of the atmosphere. The consequence of the absence of black carbon radiative effect on the simulation should be discussed.

---

## Referee Comment (RC2) · D. Baumgardner (Referee) · 5 Oct 2018

Given the reluctance of one of the reviewers to submit their comments after agreeing to do so, as the editor of this paper, I will provide the second review so that the manuscript can be revised and submitted in a timely fashion. I have attached an annotated version of the manuscript to this review where most of my comments, suggestions and corrections are posted. The primary questions that I would like addressed addressed are listed below.

This study is primarily a modeling evaluation of how mineral dust impacts the formation and evolution of cirrus clouds. The model results are compared against space-borne,

[Figure]

airborne and ground-based measurements in order to highlight how the parameterization of dust as ice nuclei (IN) impact the microphysics of of cirrus. The specifics of the model are described in detail along with sufficient references to allow the interested reader to learn more about the modeling.

I am not a modeler and hence cannot comment on most of the aspects of the simulation that are detailed. I was, however, struck by the fact that the authors wait until the final paragraphs of the discussion section to reveal that there is what I consider to be a very large discrepancy between the vertical profiles of temperature and humidity measured with the radiosondes and those simulated by the model. Given how every aspect of the modeled microphysics depend on the water vapor mixing ratio, why isn't this uncertainty introduced at the very beginning before any of the discussion of IN? I think that there has to be some type of exercise that shows the reader how sensitive the model is to these differences.

That being said, the authors suggest various reasons why the model differs from the sounding, all of them potentially valid; however, they fail to use the most powerful tool at their disposal, i.e. the aircraft that measures temperature and water vapor mixing ratio to a high degree of accuracy. If the arguments are to be convincing than I argue that the model meteorology needs comparing with that measured on the aircraft.

This brings me to my second point that concerns the dust aerosol that most certainly was being transported to the area of interest but its parameterization is based on a data base that may or may not be relevant to the study at hand. Yes, the AOD is measured with Aeronet but the only vertical profiles of dust are from a single lidar that derives an extinction coefficient from the back-scatter profiles and clearly shows two layers whereas the model only shows one. It gives little quantitative information about the aerosol microphysics. Why are the aircraft measurements not being used? Were there no aerosol spectrometers on the aircraft that could be used to estimate the dust concentrations and size distributions, as well as measuring the interstitial aerosol to see if indeed there were potentially more IN on April 4 than on April 3? The vertical

profile of the aerosols would also be provide by the aircraft.

Without these questions being addressed (as well as others in the manuscript), the final conclusions are less than impressive and we are all left wondering if it meteorology or aerosol composition that really matters.

Please also note the supplement to this comment:
https://www.atmos-chem-phys-discuss.net/acp-2018-685/acp-2018-685-RC2-supplement.pdf

[Figure]

**Supplement:**

[revised manuscript text omitted]

---

## Author Comment (AC1) · 15 Nov 2018

**Authors' Response to Reviewer's Comments**

Manuscript No.:  acp-2018-685, submitted to APC
Title:            The impact of mineral dust on cloud formation during the Saharan dust
                  event in April 2014 over Europe
Authors:          M.Weger, B.Heinold, C.Engler, et al.

*We would like to thank the reviewer for her/his time and constructive comments, and hope that we have responded satisfactorily to all the points raised.*

**Anonymous Referee #1**

**General comment**

The paper discusses the sensitivity of an upper-level cloud cover to two different microphysics schemes, with and without dust-cloud and dust-radiation feedbacks during a dust outbreak over Europe. It presents a comprehensive comparison between simulations and remote sensing observations of aerosol and cloud properties as well as in situ measurements from an aircraft campaign. The simulation with the dust-cloud and dust- radiation feedbacks provides the best results. This is attributed to enhanced deposition freezing. Different empirical ice nucleation parameterizations are then tested, which shows the importance of remaining uncertainties in the ice nucleating properties of mineral dust. Last, the best simulation is shown to be too dry in the upper troposphere, which is likely the main cause of underestimating the cloud cover. My suggestion at the end of the reading would have been to redo the work with more realistic values of specific humidity at the initial and boundary conditions. This would however be too much work and as stated in the text, it is beyond the scope of the paper. Despite this disappointing result on cloud cover prediction, the paper presents an in-depth discussion on the impact of dust on cloud cover. As such the paper deserves publication to ACP.

We thank the referee for the comment and the suggestions. Indeed, it would not be feasible, to redo all the model runs with adjusted meteorological fields. However, to get an upper proxy of the possible impact of humidity on cloud formation, we arbitrarily raised humidity in the boundary fields. The outcome of this sensitivity study is summarized in the comments below.

**Minor comments**

Page 3, line 56: Extra "is" between "homogeneous" and "nucleation".

Corrected

Page 4, line 104: Typo on "microphyiscs".

Corrected

Page 19, line 587: The lidar measurements show large values of extinction coefficient around 2-km altitude. You implicitly attribute this signal to black carbon aerosol that can absorb visible radiation. This radiative effect is not present in the simulation. Because it is a strong signal, it might have a big impact on the stability of the atmosphere. The consequence of the absence of black carbon radiative effect on the simulation should be discussed.

We complemented the discussion of dust-radiative effects on precipitation formation by the following two sentences, to discuss the absence of the black carbon layer in the model: "The layer of strongly absorbing black carbon aerosol at 2 km altitude (see fig. 6), which is not represented in the simulations, may have an enforcing effect on thermal stratification, possibly explaining some of the remaining deviations in the model."

**Referee D. Baumgardner**

We thank Darrel Baumgardner for his effort to contribute a timely review as Editor by his own and for his thoughtful and valuable review comments.

**General comment**

Given the reluctance of one of the reviewers to submit their comments after agreeing to do so, as the editor of this paper, I will provide the second review so that the manuscript can be revised and submitted in a timely fashion. I have attached an annotated version of the manuscript to this review where most of my comments, suggestions and corrections are posted. The primary questions that I would like addressed are listed below.

This study is primarily a modeling evaluation of how mineral dust impacts the formation and evolution of cirrus clouds. The model results are compared against space-borne, airborne and ground-based measurements in order to highlight how the parameterization of dust as ice nuclei (IN) impact the microphysics of cirrus. The specifics of the model are described in detail along with sufficient references to allow the interested reader to learn more about the modeling. I am not a modeler and hence cannot comment on most of the aspects of the simulation that are detailed. I was, however, struck by the fact that the authors wait until the final paragraphs of the discussion section to reveal that there is what I consider to be a very large discrepancy between the vertical profiles of temperature and humidity measured with the radiosondes and those simulated by the model. Given how every aspect of the modeled microphysics depend on the water vapor mixing ratio, why isn't this uncertainty introduced at the very beginning before any of the discussion of IN? I think that there has to be some type of exercise that shows the reader how sensitive the model is to these differences.

We agree therein, that a discussion of humidity should be given more room and priority. Therefor we moved the paragraph to the front, right after the discussion of cloud presentation in the reference model run SMBLK. In addition to radiosonde data, we also evaluated airborne AIMS-H2O measurements carried out on 3 April afternoon. Motivated by a model evaluation with these data, which shows modeled upper tropospheric humidity being dramatically too low during the second half of the simulation period, we performed a small sensitivity study, to rule out that the large discrepancies seen in the satellite retrieved cloud cover are solely due to a lack of humidity in the model. This was achieved by re-running the reference setup (SMBLK) with modified meteorological boundary fields: Between 6 km and 11 km, humidity was raised, by adding a Gaussian profile, centered at 8 km, which roughly corresponds to the maximum deviations seen in the radiosonde/model comparison. The added function is also time dependent to account for the temporal increase in measurement-model humidity difference. From 4 April 18 UTC, the added humidity is up to plus 70 % of the initial value. With this modification, cloud cover increased significantly, as to expect. It is 50 % higher, than in the reference model run without raised humidity. This, however, occurs mainly at the boundaries of the domain, and not in the center. Most likely, the humidity is rapidly absorbed by the already present ice particles, making them settle even faster. As a result, cloud dissipation is probably accelerated, as hinted by the sharp gradient in cloud cover. Anyway, given the large initial underestimation of cloud cover, the increase seen with the increased humidity values cannot decisively improve the situation, which answers the initial question and leads to the discussion of INPs. In this regard, mineral dust likely has an enhancing effect on cloud life time, as it produces many small ice particles, which tend to stabilize cloudiness.

That being said, the authors suggest various reasons why the model differs from the sounding, all of them potentially valid; however, they fail to use the most powerful tool at their disposal, i.e. the aircraft that measures temperature and water vapor mixing ratio to a high degree of accuracy. If the arguments are to be convincing than I argue that the model meteorology needs comparing with that measured on the aircraft.

As stated above, we evaluated these data for 3 April, too. However, on 4 April HALO flew outside of our model domain and therefore a direct model comparison was not possible. The new data fits well into our previous radiosonde data, as it confirms, that initially the atmosphere was not too dry, but rather too humid. The lack of humidity appeared on 4 April and increased further thereafter. Concerning the uncertainties of humidity measurements for both radiosonde and AIMS-H2O, we found that for both methods these range between of approximately 5 – 10%, if the temperature is not colder than -60°C and humidity not too low. We added a short paragraph introducing the AIMs data and discussing the uncertainties of humidity measurements in Section 2.

This brings me to my second point that concerns the dust aerosol that most certainly was being transported to the area of interest but its parameterization is based on a data base that may or may not be relevant to the study at hand. Yes, the AOD is measured with Aeronet but the only vertical profiles of dust are from a single lidar that derives an extinction coefficient from the back-scatter profiles and clearly shows two layers whereas the model only shows one. It gives little quantitative information about the aerosol microphysics. Why are the aircraft measurements not being used? Were there no aerosol spectrometers on the aircraft that could be used to estimate the dust concentrations and size distributions, as well as measuring the interstitial aerosol to see if indeed there were potentially more IN on April 4 than on April 3? The vertical profile of the aerosols would also be provided by the aircraft.

For various reasons, we consider the INP-parameterization by Ullrich et al. 2017 for desert dust as one of the most accurate to date. Nevertheless, we don't know on how well it represents the specific properties of Saharan mineral dust, as it was designed to be applicable globally and to represent a wide range of mineral dust types.

We agree, that any additional aerosol measurements, especially those airborne based, could shade more light in the real distribution of INPs during our modeling study.

Firstly, Figure 15 is replaced by a new Figure, now containing the whole spectrum of NIXE-CAS measurements. The part of which is dominated by aerosol particles is clearly indicated and used as aerosol measurements. After communicating with the corresponding PIs, we included also measurements made by an optical particle counter (OPC) aboard HALO. For particles larger than 500 nm, we consider OPC counts as a good indicator for the presence of mineral dust, especially when comparing the measurements for both days. Both used data sets (NIXE-CAS and OPC) support our assumption, that significantly more desert dust was present in the upper troposphere on 4 April, as compared to 3 April, when quite pristine conditions prevailed. This implies, that indeed on 4 April more INPs were present, possibly explaining the more widespread cirrus and higher IWC values. Unfortunately, data from the mass spectrometer ALABAMA could not provide information on the cirrus residual composition, as it was only connected for a short time to the CVI inlet. Also, the lidar aboard HALO could not contribute any additional information of the aerosol distribution, as too much cloudiness was present.

**Supplement**

Page 3, Line 56: "is"

Corrected

Page 3, Line 72: What does this mean? "By a hundred fold?"

Corrected:
"Dust particle number concentrations can exceed the climatological mean value by a hundred-fold over a wide tropospheric height range during a dust event (Hande et al., 2015)."

Page 5, Line 151: Where does this number come from? Won't the subsequent activation of IN be quite sensitive to what is used here? How is this value constrained?

This value is originally used in the scheme of Seifert and Beheng (2006). We already tested the sensitivity to this parameter by decreasing it to $10^{-14}$ kg, but this had little effect on IWC, and was not mentioned in the study. A crystal mass of $10^{-12}$ kg will not affect total available water vapor at all.

Page 6, Line 176: Substantial fall velocities

MUSCAT considers dust dry deposition by gravitational settling, which depends on the size of the individual transported bins. Consequently, it is much more likely that the smaller particles will remain in the upper troposphere.

Page 7, Line 192: Spinning, Spinning

Corrected

Page 7, Line 204: By inertial & nucleation?

This sentence was misleading, as Bangert et al. (2011) did not consider interstitial aerosol inside clouds at all. They considered only aerosol activation at the cloud base. With their approach, heterogeneous ice nucleation by mineral dust is only indirectly considered (via the increase of cloud droplet number concentration due to the Twomey effect). In our approach, some particles activate, while others remain interstitial and can interact via the contact freezing mode, which is more efficient than the immersion freezing mode.

Now it is written: " In past modeling studies, aerosol scavenging by activation and aerosol processes inside clouds were usually not considered (e.g., Bangert et al., 2011).  As a consequence, cloud freezing had to be treated stochastically only depending on cloud droplet number concentrations but not on a variable aerosol concentration (Bangert et al.,2012). Field studies, however, have shown a variable interstitial aerosol fraction, increasing toward the cloud edges (Gillani et al., 1995) or in the presence of ice particles (Verheggen et al., 2007)."

Page 7, Line 213:  Surely inorganic aerosols are important. Why are they ignored?

Part of the inorganic fraction is soot, which is included. Other than that, the parameterization of Phillips et al. (2008) considers inorganic metallic particles, which are included together with the mineral dust under "dust and metallic" in their parameterization. Assuming, however, that in our case mineral dust is much more important than metallic particles, we renamed this class to "mineral dust". This decision has also practical reasons, as the interactive aerosol simulation was only carried out for desert dust.

Page 7, Line 213: Confusing. "Latter" refers to what? What is the former?

Corrected: "the latter two classes"

Page 7, Line 215: "In the latter"

Corrected

Page 8, Line 230: Define Singular Hypothesis

We added: "Heterogeneous ice nucleation in our model is based on empirical parameterizations of the aerosol surface density of ice nucleation active sites (INAS) $n_{IS}$ [$\mu m^{-2}$], presuming the validity of the singular hypothesis, which assumes instantaneous ice nucleation events occurring in response to a sufficient increase in supersaturation, as opposed to a more detailed stochastic ice nucleation model (see e.g. Niedermeier et al., 2011)"

Page 8, Line 234: "Considered"

Corrected

Page 9, Line 257: Modeled or measured, i.e. using gridded EMWF met fields or generated by the model?

This is the grid-scale vertical temperature gradient generated by the model.

Page 11, Line 336: Concentration as well as vertical transport. How does dust get to cirrus altitudes? How much fall out and dispersion? If by convection, this will be highly inhomogeneous and wouldn't be considered a homogeneous layer

See comment on Page 17, Line 508.

Page 11, Line 349: This sentence seems contradictory.

This was not clear enough.
Now it is written: "The horizontal resolution of domain D2 is high enough to resolve moist deep convection. Nevertheless, sub-gid scale shallow convection needs to be parameterized by the restricted application of the scheme on his type of convection only, which is a common approach at this scale.

Page 12, Table 2: Surface values? How do these translate to cirrus altitude. Shouldn't these be mixing ratios rather than volume concentrations?

We agree, that indeed the mixing ratio should be kept constant across the height range, as there would be dilution of the dust plume by adiabatic expansion. The used value is adopted from Phillips et al. (2008), measured as the aerosol background concentration atop Mount Werner, at 3200m above sea level. Assuming, that the air density is by a factor of 2 or 3 lower in the upper troposphere, our assumed climatological dust concentration is too high by this factor. However, this does not change the principal outcome of our sensitivity study, as the spread between simulated dust concentrations and climatological mean concentration would be even larger.

Page 13, Line 381: Vertical resolution?

It is the horizontal resolution.

Page 13, Line 393: "Is there information from active sensors on dust layer height and depth? Any in situ measurements that would validate assumption that the dust gets to cirrus altitudes?"

We included aerosol OPC measurements here, to evaluate the vertical distribution of mineral dust.

Page 15, Line 438: "Wouldn't measurements of particles < 3 um be useful for looking at potential dust?"
Page 25, Line 770: "Compare the < 1 um concentrations in CAS for the two days to see if these correspond to the simulated values. The aircraft also has aerosol measurements, why are they not used to validate the dust layers and concentrations?"

We now use the NIXE-CAS (d<3 μm), as well as the OPC measurements from the aircraft.

Page 15, Line 438: I don't understand what this means.

The indicated sentence is removed in the context of the revision of this section.

Page 19, Line 591: Lifting by the cold front isn't simulated?

Yes, it is. We slightly modified the sentence, to make it clear: "COSMO-MUSCAT, on the other hand, only considers aeolian mineral dust."

Page 20, Line 625: "They are in disagreement. How can this be called confirmation?"

It did not refer to the agreement between model data and observations, but to the two comparisons leading to the same outcome.
Now it is clear: "In summary, these results support the differences seen in the infrared image comparison, as both comparisons implicate a substantial lack of IWC inside cirrus clouds."

Page 21, Line 646: Need a figure similar to Fig. 7 where satellite data is used as the reference.
Page 21, Line 656: Not a good way to compare data sets. Too qualitative. Why not show a figure like 7?

Now the image series of simulated infrared temperatures is complemented by the satellite images.

Page 22, Line 691: Is 0.5 significantly larger than 0.4?

Yes, it is. To show that, we added confidence intervals.

Page 24, Line 751: I don't think this suggestion can be substantiated. Just as wasy to say that the vertical motion is increasing supersaturation wrt to ice, maximizing growth then sublimation as air dries.

This should be possible to substantiate: "This suggests ice nucleation taking predominantly place at the cloud tops, where temperatures are the coldest, while after some

growth the larger particles settle and are therefore more likely to be found near the cloud base."

Page 24, Line 762: How can this be justified?

This is the positive Twomey effect for ice clouds: The more INPs are present, the more ice crystals will nucleate, ergo the competition for the available water vapor is larger, and the ice crystals end up being smaller. With the aerosol measurements from the aircraft evaluated now, this assumption can be justified.

Page 25, Line 779: Not significantly different given the uncertainties in ice crystal measurements with CAS.

Corrected: "This asymmetry results in DN=27 μm for the model, which is smaller than the measured value of 38 μm, but not significantly given the uncertainties of NIXE-CAS measurements."

Page 25, Line 785: Should compare IWC and number concentrations also in scatter plots, not just the average diameters. Also, area or mass weighted diameters are better to compare rather than average diameters.

There is a very low spatial correlation between clouds in the model and in reality, due to the restricted model resolution. Therefore, it is not very promising to show IWC in scatter plots. We already evaluated mean IWC values, and found these to be too low in the model. At this point, however, we are more interested in a detailed analysis of cloud microphysical properties. The two-moment scheme gives ice water mixing ratio and ice particle number mixing ratio as output, from which the grid-cell volumetric mean diameter of ice particles can be easily calculated, which per definition is mass weighted. (See Equation 12). Integrating the measured PSD and IWC gives the same quantity for aircraft measurements. To eliminate IWC as an important co-determinate, we compared only those model values, with an IWC of the same magnitude than the measured value. Furthermore, of these values, we took the closest to the aircraft position. This leads to the impressive scatter plots, which clearly show a trend with increased ice nucleation (more dust and/or more efficient INP-parameterization used). This should be close to the maximum of information, one can obtain with these data.

Page 26, Line 819: This would seem to be a potentially major factor in the comparison but why not test it?

Now tested, see comments from above.

Page 26, Line 831: Compare model RH with aircraft. Aircraft measurements are the most powerful tool you have to assess the model but the measurements seem underutilized.

Now used, see comments from above.